# Inter-hemispheric synchroneity of Holocene precipitation anomalies controlled by Earth's latitudinal insolation gradients

Michael Deininger [1✉], Frank McDermott[2], Francisco W. Cruz[3], Juan Pablo Bernal [4], Manfred Mudelsee[5,6], Hubert Vonhof [7], Christian Millo [8], Christoph Spötl [9], Pauline C. Treble [10,11], Robyn Pickering[12,13] & Denis Scholz[1]

Atmospheric circulation is a fundamental component of Earth's climate system, transporting energy poleward to partially offset the latitudinal imbalance in insolation. Changes in the latitudinal distribution of insolation thus force variations in atmospheric circulation, in turn altering regional hydroclimates. Here we demonstrate that regional hydroclimates controlled by the Northern Hemisphere mid-latitude storm tracks and the African and South American Monsoons changed synchronously during the last 10 kyrs. We argue that these regional hydroclimate variations are connected and reflect the adjustment of the atmospheric poleward energy transport to the evolving differential heating of the Northern and Southern Hemispheres. These results indicate that changes in latitudinal insolation gradients and associated variations in latitudinal temperature gradients exert important control on atmospheric circulation and regional hydroclimates. Since the current episode of global warming strongly affects latitudinal temperature gradients through Arctic amplification, our results can inform projections of likely inter-hemispheric precipitation changes in the future.

[1] Institute of Geosciences, Johannes Gutenberg University Mainz, J.-J.-Becher-Weg 21, 55128 Mainz, Germany. [2] UCD School of Earth Sciences, University College Dublin, Belfield, Dublin 4, Ireland. [3] Instituto de Geociências, Universidade de São Paulo, Rua do Lago 562, São Paulo, Brazil. [4] Centro de Geociencias, Universidad Nacional Autónoma de México, Campus UNAM –Juriquilla, 76230 Querétaro, Mexico. [5] Climate Risk Analysis, Kreuzstrasse 27, Heckenbeck, 37581 Bad Gandersheim, Germany. [6] Alfred Wegener Institute Helmholtz Centre for Polar and Marine Research, Bussestrasse 24, 27570 Bremerhaven, Germany. [7] Max-Planck-Institute for Chemistry, Hahn-Meitner-Weg 1, 55128 Mainz, Germany. [8] Instituto Oceanográfico, Universidade de São Paulo, Praça do Oceanográfico 191, São Paulo, Brazil. [9] Institute of Geology, University of Innsbruck, Innrain 52, 6020 Innsbruck, Austria. [10] Connected Waters Initiative Research Centre, UNSW Sydney, Kensington, NSW 2052, Australia. [11] ANSTO, Locked Bag 2001, Kirrawee DC, NSW 2232, Australia. [12] Department of Geological Sciences, University of Cape Town, University Avenue, Rondebosch 7701, South Africa. [13] Human Evolution Research Institute, University of Cape Town, Rondebosch 7701, South Africa. ✉email: michael.deininger@uni-mainz.de

Earth's atmospheric circulation redistributes energy from the low-latitudes (tropics) to the high-latitudes (extratropics) to partially compensate for the global latitudinal temperature gradients that arise from the unequal latitudinal distribution of top-of-atmosphere incident solar radiation (insolation). Changes in the latitudinal insolation gradients and associated hemispheric and inter-hemispheric temperature gradients control the strength and position of the mid-latitude storm tracks[1–5], the latitudinal positions of the Hadley cell termini[6,7], Hadley cell circulation[8,9], and the intertropical convergence zone (ITCZ)[9–12]. An important consequence of these relationships was demonstrated recently for the Holocene[3], namely that precipitation in the Northern Hemisphere mid-latitudes (30°N to 50°N) increased in response to increasing temperature gradients between the low-latitudes and the high-latitudes of the Northern Hemisphere during the past 10 kyrs. Past variations in the extratropical and tropical atmospheric circulation systems are, however, often studied separately and are generally interpreted in terms of changes in local insolation, rather than as part of a broader response of atmospheric circulation to changing latitudinal insolation and temperature gradients on hemispheric scales (see e.g., refs. [13–15]).

In the Atlantic sector (Fig. 1), variations in extratropical and tropical atmospheric circulation systems control precipitation linked to mid-latitude storm tracks and the African and South American Monsoons. As required by energy balance models[5,8,9], changing latitudinal insolation and temperature gradients during the Holocene should therefore modulate the tropical atmospheric circulation, in turn affecting tropical precipitation in Africa and South America. To adjust the poleward energy transport, these models imply increased tropical precipitation when the temperature gradient between the low-latitudes and high-latitudes weakens because of high-latitude warming. Similarly, the ITCZ shifts towards the warmer hemisphere to adjust the cross-equatorial energy transport[10]. Furthermore, it is expected that precipitation in the mid-latitudes should decrease due to weaker storm tracks[3], when the low- to high-latitude temperature gradient weakens. Since latitudinal temperature gradients are currently decreasing due to Arctic warming[16], it is important to test the hypothesis that patterns of tropical and extratropical precipitation are linked consistently to latitudinal temperature

gradients during the Holocene. If this hypothesis is proven to be robust, it can contribute to the development of an improved energy-budget framework for tropical atmospheric circulation and regional monsoon systems[17].

By combining independent hydroclimate reconstructions from the Northern Hemisphere mid-latitudes[3], with new composite records from Africa and South America (Fig. 2), we detect synchronous changes in precipitation amounts or intensity (referred to as precipitation hereafter) in the Northern Hemisphere mid-latitudes and the regions affected by the African and South American Monsoons as well as precipitation in southern African during the Holocene (10 to 0 kyr BP, BP is 'before present' referring to 1950 CE; Fig. 1). We develop a conceptual framework to explain the synchroneity of these interhemispheric precipitation responses by variations in latitudinal insolation gradients, and in turn latitudinal temperature gradients, that changed substantially during the Holocene[3,4,18].

## Results

**The hydroclimate dataset.** The new composite hydroclimate records presented here are the weighted means of the long-term trends of 37 precipitation-sensitive proxy records from five different archive types (lake sediments, ocean sediments, speleothems, peat bogs, fossil vertebrate records) from Africa ($n = 25$, Supplementary Data 1) and South America ($n = 12$, Supplementary Data 2) (see 'Methods' section). Because precipitation in the tropical and southern latitudes of Africa peaks during the boreal and austral summer, respectively, two separate composite hydroclimate records were calculated: one for the African Monsoon and one for southern African precipitation, with the latter based on hydroclimate records from southeastern and southwestern Africa (Fig. 2). Hydroclimate records from Paraíso cave and Lapa Grande cave in Brazil were omitted, because precipitation at Paraíso cave occurs year-round[19] and is therefore not solely related to the South American Monsoon, while precipitation at Lapa Grande cave is insensitive to Holocene orbital-scale changes of the South American Monsoon[20]. Seasonal and annual latitudinal insolation gradients were calculated from monthly means[21], using December to February for the austral summer (boreal winter) and June to August for the boreal summer (austral

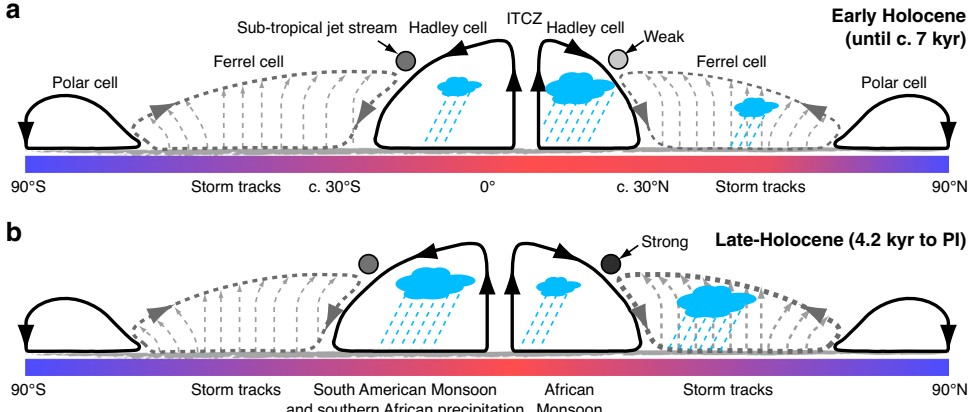

**Fig. 1 Schematic illustration of the general circulation of the atmosphere for early and late Holocene latitudinal insolation gradients. a** Schematic representation of the mean early Holocene atmospheric circulation (until c. 7 kyr) in the Atlantic sector, showing the latitudinal positions of the monsoon regions, mid-latitude storm tracks and the subtropical jet stream. During this period, the Northern Hemisphere extra-tropics warmed more and received more insolation relative to the Southern Hemisphere. This induced higher precipitation in the African Monsoon region, a northward migration of the intertropical convergence zone (ITCZ) and lower precipitation in the Northern Hemisphere mid-latitudes and in the realm of the South American Monsoon. **b** Same as in **a** but for the late Holocene (after c. 4.2 kyr) until pre-industrial (PI) times. During this period, a stronger temperature gradient between the low-latitudes and high-latitudes in the Northern Hemisphere, as well as changing latitudinal insolation gradients induced increased precipitation in the mid-latitudes and in the realm of the South American Monsoon, while precipitation decreased in the African Monsoon region.

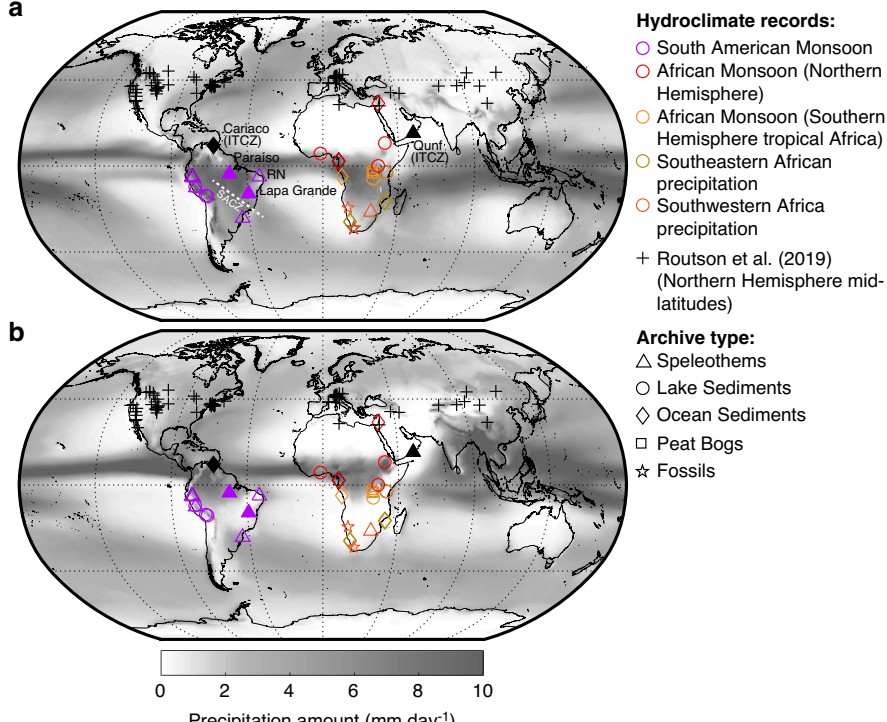

**Fig. 2 Spatial distribution of the Holocene hydroclimate records utilised in this study.** The symbols indicate the location and archive type of individual hydroclimate proxy records, including the Northern Hemisphere mid-latitude hydroclimate proxy records of ref. [3]. Filled symbols indicate locations, where the record was omitted for the calculation of composite records (see 'Methods' section). **a** Mean precipitation amount from December to March and **b** from June to September calculated from the ERA-Interim Reanalysis dataset using the years 1979–2015[49]. Each site is listed in Supplementary Data 1 and 2, including the individual hydroclimate dataset used in this study and related references.

winter). The Holocene latitudinal temperature gradients were taken from refs. [3,18], and are based on mean annual gradients. We focus here on the mid to late Holocene transition (after circa 7 kyr BP) when insolation and temperature gradients between the low-latitudes and the high-latitudes changed substantially[3,4,18] and the effects of remnant Northern Hemisphere ice sheets on the Northern Hemisphere mid-latitude atmospheric circulation had waned[3,22,23].

**Hydroclimate changes in South America.** The South American Monsoon (Fig. 3d) was weakest in the early Holocene (10–9 kyr BP), before progressively strengthening until its Holocene maximum shortly after 3 kyr BP. When the South American Monsoon was gaining strength (9–3 ka; Fig. 3d), precipitation increased on the eastern side of the Andes and in southeastern Brazil (Fig. 2a), decreased at Rio Grande do Norte[14] (RN) (Fig. 3e), and was largely unaffected at Lapa Grande cave[20] (LG) (Fig. 3f). In parallel with the intensification of the South American Monsoon, precipitation decreased in northern South America (Fig. 3b), inferred from marine sediments from the Cariaco Basin[24]. Reconstructed precipitation at Paraíso cave[19] (Fig. 3c) (eastern Amazon region) increased from 10 to 6 kyr BP, and decreased since 4 kyr BP. Thus, precipitation changes at Paraíso cave are not consistent with the persistent changes in Holocene precipitation observed for the South American Monsoon (Fig. 3d) or northern South America (Fig. 3b).

**Hydroclimate changes in Africa.** As with the South American Monsoon (Fig. 3d), the African Monsoon (Fig. 4c) shows a persistent Holocene trend, but of opposite sign. During the past 10 kyrs, the African Monsoon was strongest in the African Humid Period[25,26] (10–5.5 kyr BP), after which it weakened through to

present. These hydroclimatic changes are also observed when only compilations of hydroclimate proxy records from the Northern and Southern Hemisphere tropics (Fig. 2a) are considered (Supplementary Fig. 1). Furthermore, precipitation decreased in southern Oman[27] through the Holocene (Fig. 4b) displaying a trend similar to the African Monsoon (Fig. 4c). The southern African precipitation dipole (Fig. 4d), a measure of opposite precipitation trends between southeastern (Fig. 4f) and southwestern (Fig. 4e) Africa, also changed in parallel with the hydroclimate changes of the African Monsoon (Fig. 4c) over the Holocene. While precipitation decreased in southwestern Africa (Fig. 4e), precipitation increased in southeastern Africa (Fig. 3f) from 10 to 1 kyr BP. Subsequent to 1 kyr BP, the precipitation trends in southeastern and southwestern Africa reversed.

**Comparison of regional hydroclimate changes.** The NH mid-latitude net precipitation (Fig. 5b) was lowest in the early Holocene (10 to circa 8 kyr BP) and increased progressively since then. This reconstruction reflects mainly changes in mean annual precipitation, but is likely more sensitive to variability in summer precipitation, when the change in the NH latitudinal temperature gradient is largest[5]. Thus, the composite hydroclimate records for the NH mid-latitudes (Fig. 5b), Africa (Fig. 5c, d) and South America (Fig. 5e) all exhibit synchronous responses in regional-scale hydroclimates. These changes occur synchronously with declining precipitation in southern Oman (Fig. 4b), and in the northern parts of South America (Fig. 3b).

**Comparison of hydroclimate and insolation changes.** Variations in the magnitude of the temperature difference as well as the temperature gradient between tropical and Northern Hemisphere extratropical regions changed in tandem with the annual

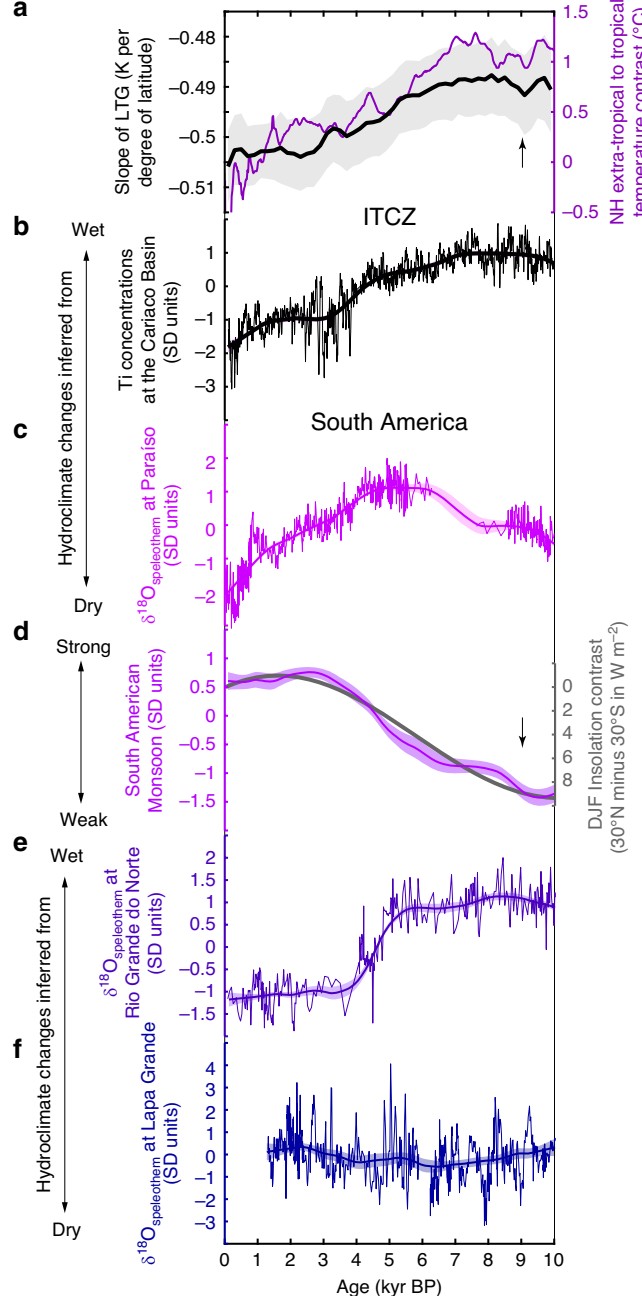

**Fig. 3 Holocene changes in the realm of the South American Monsoon.**
**a** Reconstructed slope of the Northern Hemisphere temperature gradient (LTG) (black, see ref. [5]) between the low-latitudes and high-latitudes and the temperature contrast (purple, see ref. [18]) calculated from zonal averages for the tropical region (30°S to 30°N) and the Northern Hemisphere extra-tropics (>30°N). **b** standardised precipitation changes associated with shifts of the intertropical convergence zone (ITCZ) inferred from the Ti-concentration in ocean sediments from the Cariaco Basin (Venezuela)[27]. The thin line is the original standardised proxy time series, while the thick line is the smoothed standardised proxy time series, using the same kernel method as for the calculation of the composite hydroclimate records (see 'Methods' section). **c** standardised precipitation changes in the eastern Amazon inferred from a speleothem δ18O record from Paraíso cave. **d** Standardised precipitation changes associated with the South American Monsoon (orange) and the mean interhemispheric insolation contrast between 30°N and 30°S for December to February (DJF) (bold grey line). Changes in the insolation contrast are given relative to present-day. Standardised precipitation changes inferred from speleothem δ18O values from, **e** Rio Grande do Norte[16] (RN) and, **f** Lapa Grande cave[24] (LG). Arrows point in the direction of increasing insolation quantities or an increasing slope. The thin lines in **b**, **c**, **e**, and **f** are the original standardised proxy time series, while the thick lines are the smoothed standardised proxy time series, using the same kernel method as for the calculation of the composite hydroclimate records (see 'Methods' section). Shading indicates the 1-sigma standard deviation and SD refers to standardised units.

(December–March) interhemispheric insolation contrast decreases and the latitudinal temperature gradient between the low-latitudes and the high-latitudes in the Northern Hemisphere strengthens, precipitation increases in the realm of the South American Monsoon. Precipitation also increases in the Northern Hemisphere mid-latitudes (Fig. 5b) when the latitudinal temperature gradient in that hemisphere becomes stronger, and the insolation contrast between the high-latitudes and the low-latitudes of the Northern Hemisphere increases.

## Discussion

These results demonstrate synchronous changes in precipitation, linked to variations in the strength of the African and South American Monsoons and the Northern Hemisphere mid-latitude storm tracks throughout the Holocene (Fig. 5). Because of the relatively constant Atlantic meridional overturning circulation during the Holocene[28], the observed inter-hemispheric pre-cipitation changes can only reasonably be explained by insolation variations. The Northern Hemisphere mid-latitude precipitation change since 10 kyr BP is controlled by the strength of mid-latitude storm tracks, which are in turn forced by the latitudinal temperature gradient between the low-latitudes and the high-latitudes[3]. In response to the latitudinal distribution of insolation, the latitudinal temperature gradient is controlled mainly by the Northern Hemisphere high-latitude warming, as tropical tem-peratures remained almost constant during the Holocene[3,18]. Thus, when the Northern Hemisphere high-latitudes received more insolation in the period from 10 to 6 kyr BP (Fig. 5b), the latitudinal temperature gradient was weaker (Fig. 5a), resulting in weaker storm tracks, a weakened subtropical jet stream, and reduced precipitation in the Northern Hemisphere mid-latitudes (Fig. 5b).

While the Northern Hemisphere mid-latitude precipitation shows a coherent response to Holocene insolation forcing[3], the response of local tropical precipitation to the insolation-forced Holocene hydroclimate changes in Africa and South American is not uniform (Figs. 3 and 4). The spatial pattern of precipitation

insolation contrast between the low-latitudes and the high-latitudes (Fig. 5b). Importantly, when the northern latitudes received more insolation relative to the southern latitudes (Fig. 5c, e), the latitudinal temperature gradient within the Northern Hemisphere was weaker, as the Northern Hemisphere extra-tropics, and the Arctic in particular, warmed more than tropical regions[3,18] (Fig. 5b).

Temporal changes of the new composite hydroclimate records track the changes of the respective hemispheric, and interhemi-spheric latitudinal insolation contrasts and the temperature gradients (Fig. 5). Precipitation increases in the area of influence of the African Monsoon (Fig. 5c) when the mean June–August (boreal summer) interhemispheric insolation contrast increases, and the latitudinal temperature gradient between the low- and the high-latitudes of the Northern Hemisphere is weaker. The opposite relationship is observed for the South American Monsoon (Fig. 5e). Thus, when the mean austral summer

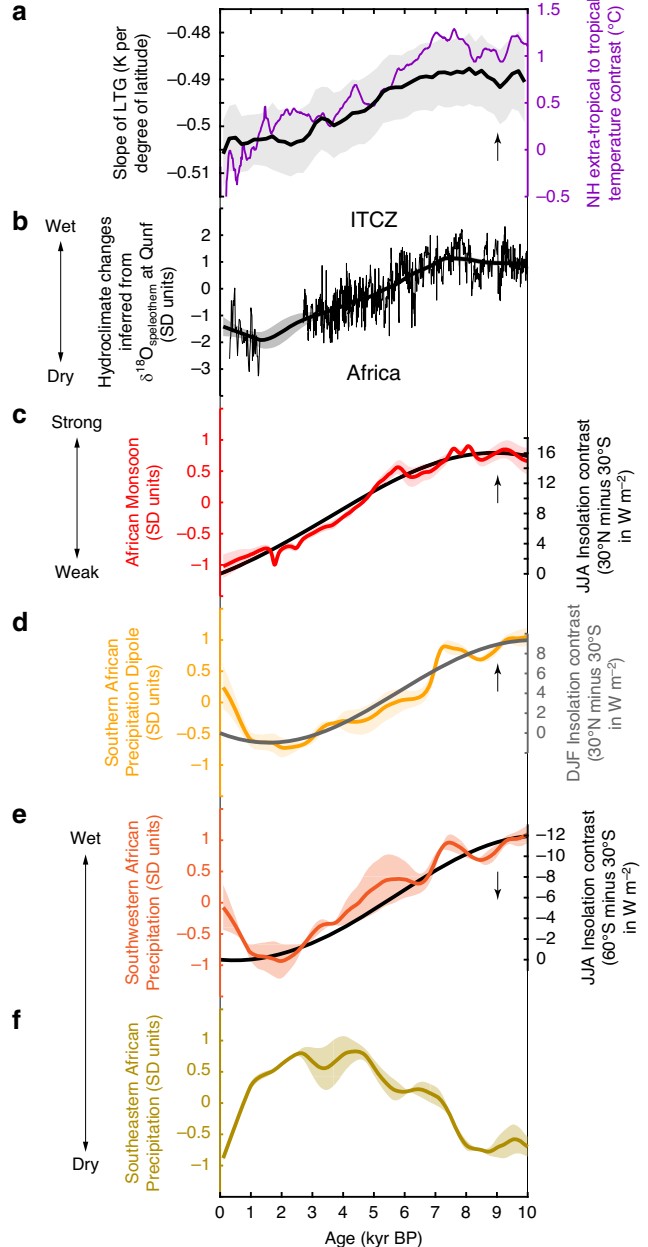

**Fig. 4 African changes of the Holocene hydroclimate. a** Reconstructed slope of the Northern Hemisphere temperature gradient (LTG) (black, see ref. [5]) between the low-latitudes and high-latitudes and the temperature contrast (purple, see ref. [18]) calculated from zonal averages for the tropical region (30°S to 30°N) and the Northern Hemisphere extra-tropics (>30°N). **b** standardised inferred precipitation changes that are associated with shifts of the intertropical convergence zone (ITCZ) inferred from a speleothem δ18O record from Qunf cave (Oman)[27]. The thin line is the original standardised proxy time series, while the thick line is the smoothed standardised proxy time series, using the same kernel method as for the calculation of the composite hydroclimate records (see 'Methods' section). **c** Standardised precipitation changes associated with the African Monsoon (red) and the mean interhemispheric insolation contrast between 30°N and 30°S for June to August (JJA) (bold grey line). Changes in the insolation contrast are given relative to present-day. A positive insolation contrast indicates that the Northern Hemisphere received more insolation than the Southern Hemisphere relative to present-day. **d** Standardised mode of southern African precipitation changes (yellow) and the mean interhemispheric insolation contrast between 30°N and 30°S for (bold grey line) during austral summer (December to February, DJF). Changes in the insolation contrast are given relative to present-day. **e** Standardised precipitation changes in southwestern Africa. **f** Standardised precipitation changes in southeastern Africa. Arrows point in the direction of increasing insolation quantities or an increasing slope. Shading indicates the 1-sigma standard deviation.

the South American Monsoon, precipitation falls nearly year-round at Paraíso cave[19] (Fig. 2). Thus, we suggest that the observed change in precipitation at Paraíso cave since 10 kyr BP is related to variations in the seasonal distribution of precipitation. From 10 to 6 kyr BP, increased precipitation was caused by a strengthening of the South American Monsoon (Fig. 3d) during the austral summer, while precipitation during the boreal summer did not change (Fig. 3b). Since 4 kyr BP, declining precipitation at Paraíso cave reflects a drying in the northern regions of South America (Fig. 3b) during the boreal summer, which outcompeted the simultaneous increase in precipitation due to strengthening of the South American Monsoon at this time. The interplay of these different seasonal hydroclimate changes yields a reasonable explanation for the complex precipitation pattern across the domain of the South American Monsoon. Ultimately, however, these tropical precipitation changes are all driven by the same insolation forcing.

When the African Monsoon was strongest, records from the Sahara and Sahel lakes[30] as well as dust fluxes from the coast west of the Sahara[25,31] indicate a wetter climate in northern Africa. Considering that the African Monsoon exerted its influence well into the Southern Hemisphere tropical region (Supplementary Fig. 1), these results support the notion of a widening of the monsoon region in parallel with increasing monsoon strength[17]. However, the composite record from the southeastern African summer rainfall zone (Fig. 4f) reveals that the seasonal influence of the African Monsoon on the summer precipitation waned within the tropical belt of southern Africa. This is because summer precipitation in the region affected by the African Monsoon (Fig. 4c) occurs during boreal summer, whereas summer precipitation in southeastern African falls during austral summer. Therefore, the opposite precipitation trends of the African Monsoon (Fig. 4c) and southeastern African summer precipitation (Fig. 4f) during the Holocene is related to the response of the boreal, and austral summer precipitation to insolation forcing. The opposing precipitation trends in southeastern and southwestern Africa are also likely to be a reflection of the seasonal distribution of precipitation, because hydroclimate records from

changes across the domain of the South American Monsoon (Fig. 3) is related to a strengthening of the lower atmospheric circulation[14], interpreted as a shift of the South American Convergence Zone (Fig. 2a), a convective band extending from the core monsoon region to the western South Atlantic[29], and strengthening of the low-level jet on the eastern side of the Andes, transporting moisture from the core monsoon region towards southeastern Brazil. This scenario can explain why precipitation decreased in Rio Grande do Norte[14] (Fig. 3e), while apparently remaining unaffected at Lapa Grande cave[20] (Fig. 3f), and while the South American Monsoon intensified at the same time (Fig. 3d). In parallel with these orbital-scale changes of the South American Monsoon, the ITCZ shifted southward in the Atlantic sector, inferred from decreasing precipitation in the northern areas of South America[24] (Fig. 3b). The persistent changes in the South American Monsoon and the migration of the ITCZ, when combined, may explain the precipitation behaviour at Paraíso cave (Fig. 3c). Unlike the other investigated regions in South America, where precipitation is mainly related to the strength of

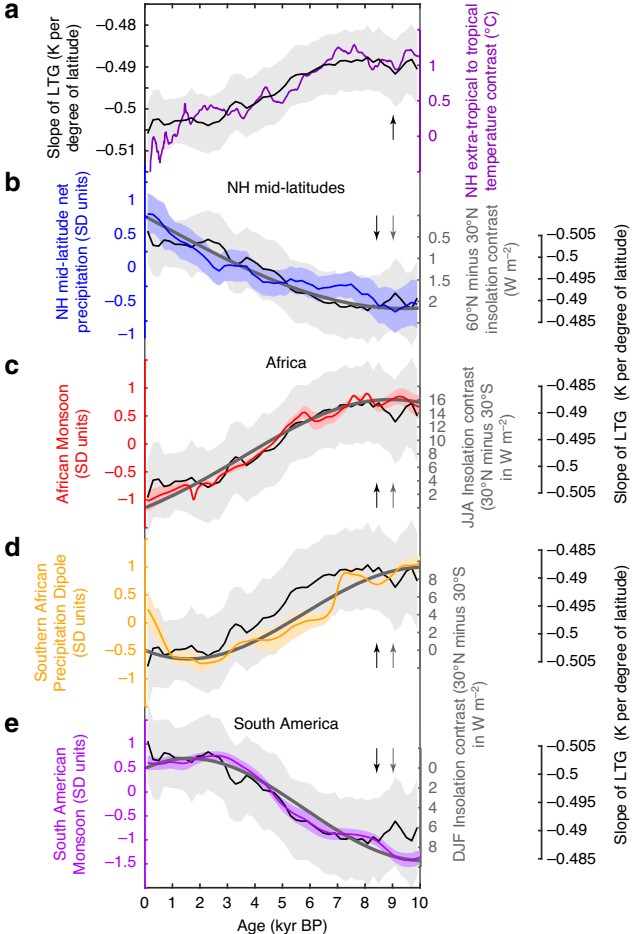

**Fig. 5 Comparison of Holocene inter-hemispheric hydroclimate variations with changes in the latitudinal energy balance.** In all panels, changes in insolation contrast are given relative to present and arrows indicate the direction of an increasing insolation contrast or a weaker Northern Hemisphere latitudinal temperature gradient. **a** Reconstructed slope of the Northern Hemisphere latitudinal temperature gradient (LTG) (black, see ref. [5]) between the low-latitudes and high-latitudes, and the temperature contrast (purple, see ref. [18]) calculated from zonal averages for the tropical region (low-latitudes) (30°S to 30°N) and the Northern Hemisphere extratropics (>30°N). **b** Standardised Northern Hemisphere (NH) mid-latitude net precipitation (blue) reconstructed by see ref. [3] and the annual mean insolation contrast between 60°N and 30°N calculated from mean monthly insolation values (bold grey line). A positive insolation contrast indicates that the Northern Hemisphere higher latitudes received more insolation than the Northern Hemisphere lower latitudes relative to present. **c** Standardised precipitation changes associated with the African Monsoon (red) and the mean interhemispheric insolation contrast between 30°N and 30°S for June to August (JJA) (bold grey line). A positive insolation contrast indicates that Northern Hemisphere received more insolation than the Southern Hemisphere compared to present. **d** Standardised mode of southern African precipitation changes (yellow) and the mean interhemispheric insolation contrast between 30°N and 30°S for December to February (DJF) (bold grey line). **e** Standardised precipitation changes associated with the South American Monsoon (orange) and the mean interhemispheric insolation contrast between 30°N and 30°S for December to February (DJF) (bold grey line). The black line illustrates the slope of the Northern Hemisphere latitudinal temperature gradient illustrated in **a**. Shading in panels a-e indicates 1-sigma standard deviation (see 'Methods' section).

southeastern Africa are sensitive to variations in summer precipitation (summer rainfall zone), while those from southwestern Africa are from regions where most precipitation falls during austral winter (winter rainfall zone). During austral winter, the insolation gradient between the low-latitudes and the high-latitudes of the Southern Hemisphere was stronger in the early Holocene. Using the same arguments as for Northern Hemisphere storm tracks[3] and assuming that the latitudinal temperature gradient between the low-latitudes and the high-latitudes in the Southern Hemisphere is forced by the latitudinal insolation gradient in the Southern Hemisphere, the Southern Hemisphere mid-latitude storm tracks were probably stronger during the early Holocene, weakening gradually during the Holocene. This would impact austral winter precipitation in southwestern Africa. The opposite Holocene precipitation trends observed in southeastern and southwestern Africa are therefore interpreted to be related to changing latitudinal insolation gradients at this time. In summary, the non-uniform regional precipitation pattern in Africa and South America are observed at the margins of the regions, that are under influence of the seasonally developed monsoon systems and where precipitation falls year-round, and changes in seasonality may exert influence on precipitation sensitive proxies. These regional precipitation patterns were considered when composite hydroclimate records are calculated (see 'Methods' section).

Conventionally, changes in the monsoon climate and its associated precipitation variations are interpreted in terms of regional-scale land-sea thermal gradients[17]. When the regional thermal gradient between land and ocean increases, the monsoon circulation gains strength and precipitation increases. On orbital time scales, monsoonal systems show clear precession and obliquity cycles, which are suggested to be forced by modified regional land-sea thermal gradients caused by insolation changes, as well as a high-latitude remote climate forcing linked to the Northern Hemisphere cryosphere[32]. Recent modelling suggests that the obliquity signal in the monsoon records can also be caused by changing meridional and interhemispheric insolation gradients[33,34]. The robust and mechanistically consistent relationships between past monsoon changes and variations in insolation on orbital time scales (e.g., see refs. [13–15,26,30,35–39]) support the land-sea mechanism. The hydroclimate variations of the African and South American Monsoons can be explained consistently by the conventional land-sea mechanism (Supplementary Fig. 2), indicating more vigorous African and South American Monsoons when the boreal or austral summer insolation is strong. However, the tropical circulation is a component of a planetary-scale circulation that redistributes energy from the low-latitudes to high-latitudes and, thus, the regional land-sea mechanism for monsoon circulation is likely to be too simplistic[17]. For example, the monsoon circulation is strongest when the land surface has already cooled due to increased monsoonal precipitation and cloudiness, in turn reducing the land-sea thermal gradient[17].

Following the concept of an energy-budget framework for tropical atmospheric circulation[17] and considering atmospheric circulation as a planetary-scale conveyer belt of energy and momentum, we hypothesise that changes in the Holocene tropical atmospheric circulation and related interhemispheric hydroclimate changes (Fig. 5) are linked to adjustments of the poleward atmospheric energy transport, due to the differential heating of the Northern and Southern Hemisphere. A general pattern that emerges from the Holocene relationships between the interhemispheric precipitation variations in the tropics and latitudinal insolation changes (Fig. 5), is that when summer insolation

increases (e.g., during boreal summer in the Northern Hemisphere) relative to the winter insolation (e.g., during austral winter in the Southern Hemisphere), tropical summer rainfall increases. A similar relationship is observed for latitudinal temperature changes (Fig. 5), revealing that tropical rainfall increases during boreal (Northern Hemisphere) summer, when the Northern Hemisphere temperatures increases more relative to the tropics, and the latitudinal temperature gradient between the low-latitudes and high-latitudes is weaker. Similarly, tropical rainfall during the austral summer increases when the Northern Hemisphere cools relative to the tropics, and the latitudinal temperature gradient between the low-latitudes and high latitudes becomes stronger. In detail, during the last 10 kyrs, the African Monsoon weakened (Fig. 5c), the South American Monsoon intensified (Fig. 5e) and precipitation in the summer rainfall zone of southern Africa increased (Fig. 4f) when the interhemispheric insolation contrast reduced during the boreal (Fig. 5c), as well as the austral summer (Fig. 5e). This can explain the similarity of the inferred orbital-scale precipitation changes in the summer rainfall zone (Fig. 4f) in southern Africa with variations of the South American Monsoon (Fig. 3d) as well as its opposite trend with the African Monsoon (Fig. 4c) during the Holocene. Precipitation peaked in southern Africa and the South American Monsoon during the austral summer, while the African Monsoon was most active during the boreal summer (Fig. 2). The observed inter-hemispheric changes in tropical summer precipitation in African and South American monsoons (Fig. 5c–e) are in agreement with theoretical predictions (Fig. 1) and with the inferred Holocene southward migration of the ITCZ over the Atlantic[24] (Fig. 3b) and the Indian Ocean[27] (Fig. 4b).

The tropical summer precipitation changes suggest that similar to the energy-flux framework of the ITCZ[10], variations in tropical summer precipitation are forced by the changing differential heating of the Northern and Southern Hemispheres and associated latitudinal temperature changes. Local processes, such as atmosphere-ocean and atmosphere-ecosystem feedbacks[9,14,19,40,41], are likely to modify the response of the regional monsoon systems[32,40,42,43] to external insolation forcing. Thus, variations in tropical summer precipitation and migrations of the ITCZ can be interpreted as responses of the tropical atmospheric circulation in order to adjust the tropical poleward energy transport[8–10]. Because variations in the latitudinal insolation gradients control not only the interhemispheric differential heating, but also the temperature gradients between the low-latitudes and the high-latitudes, the interhemispheric hydroclimate changes offer a paradigm for an extended energy-budget framework that connects tropical and extra-tropical atmospheric variations and associated hydroclimate changes (Fig. 5). While the mechanism that explains the relationship between the strength of mid-latitude storm tracks and latitudinal insolation and temperature gradients is well described[3], it remains unclear whether the response of tropical summer rainfall in Africa and South America to variations in latitudinal insolation and temperature gradients is forced predominantly by tropical or extratropical mechanisms, or by some combination of both. The temperature gradient between the low-latitudes and the high-latitudes not only alters the strength of mid-latitude storm tracks, but also the subtropical jet stream. The latter becomes weaker when the latitudinal temperature gradient weakens[3]. This may affect the momentum-transport between the extratropical and tropical atmospheric circulation, which modulates the vigour of the Hadley circulation, particularly at the beginning of the monsoon season[10,17]. Thus, momentum transport may increase when the jet stream weakens in response to a weaker latitudinal temperature gradient. Our key argument is that the tropical poleward and cross-equatorial energy transport is altered by

variations in atmospheric circulation that are in turn forced by changes in the latitudinal distribution of insolation and latitudinal temperature gradients[5,9,10,33,34]. If one hemisphere receives more insolation ('warm' hemisphere) relative to the other ('cool' hemisphere), the tropical poleward energy transport would be reduced in the warm hemisphere, while the cross-equatorial energy transport from the warm to the cool hemisphere would increase the poleward energy transport in the cool hemisphere at the same time[10,11]. However, if changes in the cross-equatorial energy flux cannot fully dissipate the 'available' excess energy in the warmer tropical hemisphere, this hemisphere will warm-up. This will likely further strengthen the tropical atmospheric circulation, especially over the continents, where heat capacity is lower than over the ocean. This process is similar to the traditional insolation-based mechanism in strengthening the land-sea thermal gradient when insolation increases, but would be strongest in late summer, when the temperature gradient between the low-latitudes and the high-latitudes is weakest. As a result, monsoon systems will be strengthened due to an increasing convective energy inducing greater precipitation. The almost constant tropical temperatures during the Holocene[3,18] could thus be a result of the cooling or dampening influence of tropical monsoon systems.

Our study reveals synchronous precipitation changes in the Northern Hemisphere mid-latitudes, Africa and South America during the Holocene. We argue that these regional hydroclimate variations are connected and mirror the Holocene variations in the extratropical and tropical atmospheric circulation, that must occur to adjust the poleward energy transport to the changing differential heating of the Northern and Southern Hemisphere. While the need for an improved and unified theory for regional monsoon systems is clear[17], our study suggests that such a theory should consider the effects of differential heating of the Northern and Southern Hemispheres and the role of the extratropics as a driver for tropical atmospheric circulation. Recent increases in atmospheric $CO_2$ concentration may cause nonstationary effects (see e.g., refs. [5,6,44,45]) that could limit the reliability of projections based on the 'early and mid-Holocene' analogue of this study. Nevertheless, our study emphasises the importance of latitudinal temperature gradients for a deeper understanding of past, present and future atmospheric variability in the troposphere and associated changes in the near-surface climate. Furthermore, in a future climate system involving greater warming of the Northern Hemisphere extratropics, and the Arctic in particular, than the tropics and the Southern Hemisphere extratropics, atmospheric circulation may shift to a mode akin to the early to mid-Holocene. That would result in weaker Northern Hemisphere mid-latitudes storm tracks[3], a weaker South American Monsoon, and a stronger African Monsoon, while precipitation in the summer rainfall zone of southeastern Africa would be lower.

## Methods

**Holocene hydroclimate datasets**. For the calculation of the African and South American hydroclimate composite time series, published hydroclimate proxy time series from Africa and South America were included if (i) they spanned at least the transition from the early to late Holocene (from 8 to 2 ka BP), (ii) had a mean temporal resolution of better than 600 years and (iii) a minimum of one radiometric age every 3000 years. The hydroclimate proxy time series were mainly compiled from the NOAA-WDS Palaeoclimatology and the PANGAEA data repositories (Supplementary Data 1 and 2). Furthermore, the Northern Hemisphere mid-latitude hydroclimate composite time series from ref. [3] was used, which is the most complete hydroclimate composite time series for the Northern Hemisphere mid-latitudes currently available in terms of spatio-temporal coverage.

**Data normalisation**. Prior to the calculation of the hydroclimate composite time series, each individual hydroclimate proxy time series was normalised by subtracting its mean value and dividing by its standard deviation. Subsequently, the normalised proxy values for a time series were multiplied by –1, if decreasing proxy

values indicate the increasing precipitation amounts. Each normalised time series thus possesses zero mean and unit standard deviation, and increasing values indicate increasing precipitation amounts. This assures that inferred hydroclimate changes between different records can be compared meaningfully to each other and that a composite hydroclimate time series can be calculated, even if the time intervals covered by the individual series are different from each other.

**Trend analyses.** The trend analysis was performed in a nonparametric manner (kernel technique) to allow for flexibility of the trend shape (not parametrically restricted). We employed the Gasser–Müller kernel[46,47] with a parabolic kernel function of a certain bandwidth. This technique has the advantages that (i) the bias at the time interval boundaries is corrected for by means of modified kernels, and (ii) the trend estimate can be calculated at prescribed time values. These time values, $t_j$, were set from 0 to 10 ka BP at a spacing of 1 year. The kernel bandwidth was set equal to 1000 years to inspect long-term trends over the Holocene. The resulting trend estimates are denoted as $X_i(t_j)$, where $i$ indexes a certain time series. The kernel technique is augmented with moving-block bootstrap resampling from the residuals (defined by data minus trend estimate) in order to determine the trend estimation standard errors, $s_{Xi}(t_j)$, that are robust with respect to the distributional shape and autocorrelation[47].

**Composite calculation.** To calculate a composite hydroclimate time series from a compilation of normalised hydroclimate proxy time series, we employed the weighted mean ($X$) and its external error ($S_{ext}$) from the compilation of hydroclimate records[47,48]. For each composite time series, the trend estimate, $x_i(t_j)$, at time $t_j$ and its bootstrap standard error, $s_{Xi}(t_j)$ of $m$ hydroclimate records ($i = 1, …, m$) are considered. The weighted mean is then defined as

$$\langle X \rangle (t_j) = \left[ \sum x_i(t_j)/s_{x_i}(t_j)^2 \right] \Big/ \left[ \sum 1/s_{x_i}(t_j)^2 \right].$$

The external error of the weighted mean at times $t_j$ is defined as

$$S_{ext}(t_j) = \left\{ \sum \left[ \left( x_i(t_j) - \{X\}(t_j) \right)/s_{x_i}(t_j) \right]^2 \right\}^{1/2} \Big/ \left\{ (m-1) \left[ \sum 1/s_{x_i}(t_j)^2 \right] \right\}^{1/2}.$$

If multiple independent hydroclimate records were available from a single location, the weighted mean and its external error were calculated for this location first. This location-specific weighted mean and its external error were then used to compute the composite hydroclimate time series instead of using all hydroclimate records from this location, which may have biased the results. For the calculation of the composite hydroclimate record indicating precipitation changes associated with the African Monsoon, African hydroclimate records (15 hydroclimate records) were used that are located in the Northern Hemisphere (six hydroclimate records) and in the tropical regions of the Southern Hemisphere (nine hydroclimate records) (Fig. 2). For the composite hydroclimate records that are linked to precipitation changes in southern Africa (five hydroclimate records), hydroclimate records from southeastern (two hydroclimate records) and southwestern (three hydroclimate records) Africa were used. Because of an observed anticorrelation of reconstructed Holocene precipitation amounts in southeastern and southwestern Africa, the hydroclimate records from southeastern Africa were multiplied with −1 for the calculation of the 'southern African precipitation dipole'. For the calculation of the composite hydroclimate record indicating precipitation changes associated with the South American Monsoon all South American hydroclimate records were used except for the hydroclimate records from Paraíso Cave and Lapa Grande. The Lapa Grande record was omitted because it is located in a region where precipitation amounts did not change significantly during the Holocene (Fig. 4f), which would have biased the calculation of the composite hydroclimate record for the South American Monsoon. The record from Paraíso Cave was omitted from the calculation, because precipitation at Paraíso (eastern Amazon region) depends on the South American Monsoon during the austral summer as well as the ITCZ during austral winter (boreal summer), and is therefore not solely a reflection of changes in the strength of the South American Monsoon (see also main text). Because the hydroclimate record of Rio Grande do Norte is anticorrelated to the South American Monsoon[14], its trend estimate was multiplied with −1 for the calculation of the composite time series. In addition, hydroclimate records were not used if they are sensitive to migrations of the ITCZ (two hydroclimate records, one from the Cariaco Basin and one from southern Oman).

## Data availability

The hydrologic proxy and temperature time series are from published work and the references to the original publications are listed in Supplementary Data 1 and 2 for the hydrologic proxy time series from Africa and South America. These datasets are also made available through [https://www.pangaea.de]. The Northern Hemisphere mid-latitude precipitation time series is available at [https://www.ncdc.noaa.gov/paleo-search/study/25890] and was calculated as well as the change of the latitudinal temperature gradient by ref. [3] from published data. The original regional Holocene temperatures from ref. [18] used to calculate the latitudinal temperature contrast are available at [https://science.sciencemag.org/content/suppl/2013/03/07/339.6124.1198.DC1]. The ERA-Interim Reanalysis precipitation dataset used to create Fig. 2 is available at [https://www.ecmwf.int].

## Code availability

The code used to calculate the long-term trends of the African and South American hydrologic proxy time series in this study was developed by Manfred Mudelsee (Climate Risk Analyses, Germany) and is available at [https://www.manfredmudelsee.com/soft/kernel/index.htm]. The precipitation patterns (Fig. 2) were plotted using the function surfm.m of the MATLAB mapping toolbox and are available from Michael Deininger upon request.

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

## Acknowledgements
M.D. thanks Augusto Mangini, Norbert Frank and Yuval Burstyn for comments on an earlier version of this manuscript. M.D. acknowledges funding by the German Research Foundation (DFG) grant DE 2398/3-1. F.M.D. acknowledges funding from the Irish Research Council through grant GOIPD/2015/789.

## Author contributions
M.D. developed the concept and designed this study. M.M. carried out the kernel analysis; M.D. calculated the composite records. F.C., J.P.B., M.M., H.V., C.M., C.S., P.T., R.P., and D.S. were involved in the interpretation of the results. M.D. and F.M.D. led the writing of this manuscript with the support of all co-authors.

## Funding

## Competing interests
The authors declare no competing interests.
