## [Peer Review File · Nature Communications]

Reviewers' Comments:

Reviewer #1:

Remarks to the Author:

Deininger et al. present an interesting hypothesis and analysis connecting large-scale circulation systems including Northern Hemisphere storm tracks, the African Monsoon, and South American Monsoon to changes in latitudinal temperature gradients. I haven't completed a fine-detailed (line by line) review of the paper in light of the following more fundamental problem.

My primary concern with the manuscript is relying on three sites to argue for a grand unifying theory of Holocene circulation change. Far more than three sites are needed to characterize a change within any one of these major circulation systems. The three records are intriguingly coherent, yet the authors need to present much more evidence to indicate these sites are representative of the very broad and nuanced circulation systems they intend to characterize. Unfortunately, too many factors influence site and proxy specific variability to have confidence in the conclusions drawn here. The changes in the three records, for example are very similar to Holocene temperature change—to what extent are these records influenced directly by temperature (not just via the LTG as proposed)? Temperature could influence cave records for example through evapotranspiration limiting the amount of infiltration at cave sites and to the isotopic fractionation of precipitation. And temperature is only one of a myriad of factors that could influence site specific variability.

The days in paleoclimate of picking a handful of our favorite records have passed. There are hundreds of publically available records (many of them directly relevant to these questions) that should be drawn upon. Adding more records will inherently increase the noise, but so too, our confidence in a signal. An in-depth and nuanced treatment of the available hydroclimate records across space is needed to help unravel changes in and drivers of Holocene circulation.

Additional comments:

Fig 1 map. The delineation of precipitation seasonality versus jet stream seasonality took me some time to unravel. Perhaps using more differentiation in the color or pattern schemes would be helpful.

Fig 1 and Lines 91-93: It is difficult to assess from the map if the sites are well positioned or not. I see the justification shown in panels b and c; however, they may or may not be positioned ideally to capture latitudinal shifts in these major circulation systems. Spannagel is located above both summer and winter Jet-streams. Summer jets are relatively weak, and thus a norward shift in this system would need to be substantial to register a change. More sites are needed to assess if and how the broader systems changed.

Fig 2 and paragraph lines 131-150. Spannagel shows higher winter precipitation and lower summer precipitation during the early Holocene. On a hemispheric scale, PMIP3 models at a similar latitude show large-scale precipitation decreased during the months January-October, with the largest reductions in August. Positive November and December anomalies are very subtle in the models.

Comment: Using the balance of summer versus precipitation as a metric for storm track position is challenging for several reasons. First, the length and intensity of seasons changed through the Holocene. Second, It is also impossible to differentiate which season caused the change. Reductions in winter precipitation are not necessarily balanced by increases in summer precipitation and vis versa.

Comment: The proposed forcing mechanism, (reducing the LTG) would probably result in a weaker average jet stream rather than a marked shift in position. If anything, there might be a somewhat

equatorial shift in the circulation.

Line 147-148. The compilation in Ref 3 is assessing total effective precipitation, blending seasons, in a broad region where precipitation is primarily influenced by extratropical cyclones. Thus, it remains unclear how the general Holocene pattern of mid latitude effective precipitation (low early Holocene toward higher late Holocene) is reflected seasonally.

Lines 222-246. Connecting solar variability to forcing the high frequency variability in these records is unconvincing. Tuning and wiggle matching records within age uncertainties will always result in higher correlations and interesting patterns to discuss. It's surprising that even higher correlations were not achieved through this method.

Reviewer #2:

Remarks to the Author:

The manuscript "Persistent inter-hemispheric synchronicity of Holocene precipitation anomalies in Europe, East Africa and South America controlled by Earth's latitudinal temperature gradients" by Deininger et al. is an interesting and important contribution to the understanding of the various monsoon systems and the impact of insolation gradients on precipitation across the hemispheres. It convincingly shows that proxy paleohydrological records from three continents located at the NH, Tropics and SH. The fact that tropical climate systems (and proxy records thereof) can be influenced by what is happening in the extra-tropics is something that has not been very well understood and this manuscript shades some light on this, which in my opinion is an excellent contribution.

However, the authors need to consider the following issues concerning the data, discussion and conclusions in view of improving the quality of the manuscript.

General Comments:

1. The authors did not make it clear why these three sites were particularly selected for comparison. This leads to several other questions: how comparable are the data? Do the different proxies really respond to the same climate forcing? At what resolution does the comparison make sense? Millennial? Centennial? This is particularly important when you compare the two speleothem records with the Lake Tana record...There seems to be some growing evidence that Lake records in the tropics may not be directly comparable with speleothem records particularly during variable climate (which is particularly true for the Late Holocene).
2. What was the reason for selecting Lake Tana record for the tropics? There are several other Holocene records from the tropics with probably more resolution than Lake Tana. And there is a more recent record/or interpretation of the lake record of Lake Tana which uses Ca/Ti ratio as a proxy for wetness/dryness (rainfall or hydrological proxy) than the absolute Ti content (Lamb et al. 2018; Scientific Reports, 8). I suggest that the authors refer to this record instead.
3. The authors seem unfamiliar with the literature on the Quaternary records of tropical Africa (see specific comments for some suggestions to improve this).

Specific comments:

Line 2 (and where ever it reappears in the manuscript with the same context): "eastern Africa" is the correct term if you refer to Ethiopia (where Lake Tana is located). "East Africa" is more a political/economic block to which only Kenya, Tanzania and Uganda belong. In fact, the region where the East African Monsoon System affects is "eastern Africa", which includes also "East Africa"

Lines 96-97: The papers cited are not the appropriate ones for describing the climate system in Ethiopia. I am not prescribing any specific papers but some suggestions are the following or any similar others on the climate systems in the horn of Africa/Eastern Africa: Segele et al., 2009; Journal of Climate, 22; Segele et al., 2009, International Journal of Climatology, 29, etc.

Lines 107-109: Detrital input could also be related to short but intense rainfall within a generally dry period. I think using an elemental ratio record (such as Ca/Ti) will be better suited than an absolute Ti content which is not easy to interpret under various hydrological conditions.

Line 153 (Figure 2): The pictures are not easy to understand as there are several variables plotted (4 in panel b). It is better to organize them differently (or use very contrasting colours for the different proxy curves) in order to clearly see how the curves trend...Panel b is very busy and it is really difficult to see if there is correlation or not.

Lines 181-188: Show what is discussed here in the diagram (as panel c of figure 2, for instance) similar to the other comparisons.

Lines 212-213 (caption to Figure 3): Is there any particular reason to reverse only these? And it is not clear if the axis is different in panel a.

Lines 219-220 (caption to Figure 3): same comment as above.

Line 231: typo "Laka"

Lines 319-321: There are several works which show that the ITCZ was high up in the North in Africa during the Early to Middle Holocene [It is recommended to cite some previous works which show this was the case, as I do not believe this conclusion is new] but it is interesting to see that the same mechanism (that led to the ITCZ to hang up in the North) had more global inter-hemispheric reach and implication.

Lines 392-394: This is an important conclusion which will have a big societal implication. But the conclusion could be more justified if the authors tried to support this conclusion by what other works show (such as modeling works or even current trends from instrumental rainfall records).

Reviewer #3:

Remarks to the Author:

The authors analyze three records related to precipitation changes in Central Europe, East Africa and southeastern Brazil that display strong similarities over the Holocene. From those records, the authors argue that the changes in the North Atlantic storm tracks, the East African Monsoon System, and the South American Monsoon System are linked and controlled by latitudinal gradient of insolation and temperature.

The strong correlation between the three records is interesting and intriguing and the authors address fundamental questions about the dynamics of our climate. However, I cannot recommend the publication of the manuscript in its present form as the strong conclusions reached by the authors are not sufficiently justified on the basis of their results and of their discussion.

Two main points justify this recommendation.

1/ Three records only are selected to derive general conclusions about the climate dynamics at large-scale. However, there is nearly no discussion of the reason of the choice of those records, except that there are located in contrasted conditions. We know that precipitation records can display large variability, even at regional scale. A brief comparison with some European records is provided figure S4 but, intriguingly, the correlation between the European records seems lower than between the selected European record and the ones in East Africa and Brazil. Before using those three records for the analyses, it is necessary to determine how they compare to other records in nearby regions. It would even be better to use a large dataset and see how this influences the proposed conclusions.

2/ The authors propose a conceptual model (summarized on Figure 1) to explain the observed variations in the three records, based on some general physical principles and existing literature but the reasoning leading to this model is speculative and the proposed hypothesis are not properly justified or tested. First, the discussion is a bit loose on the concepts themselves as well as on the season analyzed. Is the figure 1 representing a zonal mean for the whole Earth (while we know that different longitudes can behave quite differently) or just for the Atlantic Sector which is closer to the data selected? Is it a figure on annual mean or for a specific season? Several hypotheses have been proposed about circulation changes in the Holocene but alternative hypotheses are not discussed at all. A large part of the argumentation is based on ref 4 but not on other work on the subject. Specifically, the authors do not explain why their conceptual model explains better the records than alternative ones. They argue for a dynamic link between the three regions but a direct link to local insolation may also explain the recorded signal (at least in some regions, for instance in the regions of monsoon as argued in some studies, e.g. line 56)? The link with the latitudinal gradient does not seem particularly convincing to me from Fig. 2. The differences before 7 ka might be due to the remnant ice sheets but they are quite large to be justified by this effect only. However, even after that, the maximum in gradient for Europe (Fig 2 a) occurs basically 1 ka later than the maximum in the precipitation records. Finally, all the records are relatively flat over the last 4000 years while the latitudinal gradient decreases clearly over this period. Without additional, more quantitative analyses, it is thus difficult to justify the strong conclusions reached by the authors.

An additional point is that I found the discussion of the solar forcing also quite speculative and it misses one important point. The response to TSI should be different from the one to orbital forcing. In particular, the effect of solar forcing should be much more homogenous between the season and the two hemispheres. This point is nearly not discussed. Furthermore, the authors did not mention solar forcing in the abstract or in the introduction so it appears disconnected to the main theme of the paper.

The authors use a lot of acronyms with which the reader is likely not familiar (and are sometimes defined after they are used such as LIG in the abstract). This makes the paper sometimes difficult to follow.

Dear Reviewers,

Thank you very much for your rigorous review, your comments and ideas. These are greatly appreciated and have prompted us to re-examine our approach which has now led to an improved manuscript and strengthened our main findings.

All reviewers shared a common concern that our interpretation was based on too few proxy records. We thank the reviewers for highlighting this and agree that this part of our approach required improving to enable a more robust analysis of our hypothesis. To address this concern we have taken the opportunity to analyse a much larger compilation of hydroclimate proxy records. This has allowed us to calculate composite records for the African and the South American Monsoon. The record selection and the applied methods are explained in detail in the revised method section of the manuscript. Furthermore, using the recently published net precipitation reconstruction for the Northern Hemisphere mid-latitudes (Routson et al., 2019), has allowed us to link variations in the African Monsoon, the South American Monsoon and in Northern Hemisphere storm tracks with changes in latitudinal insolation gradients and latitudinal temperature gradients. Overall, the new results strengthen the conclusion drawn in the previous manuscript and the revised manuscript has been updated accordingly. We think that it provides now firm and robust insights into Holocene precipitation changes in the Atlantic sector linking storm tracks in the Northern Hemisphere mid-latitudes, the African Monsoon and the South American Monsoon.

Reviewer comments have black letters
Authors response have blue letters

Reviewer #1 (Remarks to the Author):

Deininger et al. present an interesting hypothesis and analysis connecting large-scale circulation systems including Northern Hemisphere storm tracks, the African Monsoon, and South American Monsoon to changes in latitudinal temperature gradients. I haven't completed a fine-detailed (line by line) review of the paper in light of the following more fundamental problem.

My primary concern with the manuscript is relying on three sites to argue for a grand unifying theory of Holocene circulation change. Far more than three sites are needed to characterize a change within any one of these major circulation systems. The three records are intriguingly coherent, yet the authors need to present much more evidence to indicate these sites are representative of the very broad and nuanced circulation systems they intend to characterize. Unfortunately, too many factors influence site and proxy specific variability to have confidence in the conclusions drawn here. The changes in the three records, for example are very similar to Holocene temperature change—to what extent are these records influenced directly by temperature (not just via the LTG as proposed)? Temperature could influence cave records for example through evapotranspiration limiting the amount of infiltration at cave sites and to the isotopic fractionation of precipitation. And temperature is only one of a myriad of factors that could influence site specific variability.

The days in paleoclimate of picking a handful of our favorite records have passed. There are hundreds of publically available records (many of them directly relevant to these questions) that should be drawn upon. Adding more records will inherently increase the noise, but so too, our confidence in a signal. An in-depth and nuanced treatment of the available hydroclimate records across space is needed to help unravel changes in and drivers of

Holocene circulation.

We agree with the reviewer and have now invested in presenting a larger hydroclimate dataset that we analyzed for coherent changes. This has resulted in an improved manuscript with a strengthened interpretation (please see our general reply to all reviewers).

Additional comments:

Fig 1 map. The delineation of precipitation seasonality versus jet stream seasonality took me some time to unravel. Perhaps using more differentiation in the color or pattern schemes would be helpful.

Fig 1 and Lines 91-93: It is difficult to assess from the map if the sites are well positioned or not. I see the justification shown in panels b and c; however, they may or may not be positioned ideally to capture latitudinal shifts in these major circulation systems. Spannagel is located above both summer and winter Jet-streams. Summer jets are relatively weak, and thus a norward shift in this system would need to be substantial to register a change. More sites are needed to assess if and how the broader systems changed.

The substantially revised Figure 1 is intended to show the conceptual picture of the atmospheric circulation and inferred changes in the respective hydroclimates to the Holocene forcing. Figure 2 shows the precipitation patterns for the months from December to March and for June to September in two separate panels to better illustrate seasonal changes in the investigated regions, as well as the location of used hydroclimate records.

Fig 2 and paragraph lines 131-150. Spannagel shows higher winter precipitation and lower summer precipitation during the early Holocene. On a hemispheric scale, PMIP3 models at a similar latitude show large-scale precipitation decreased during the months January-October, with the largest reductions in August. Positive November and December anomalies are very subtle in the models.

Comment: Using the balance of summer versus precipitation as a metric for storm track position is challenging for several reasons. First, the length and intensity of seasons changed through the Holocene. Second, It is also impossible to differentiate which season caused the change. Reductions in winter precipitation are not necessarily balanced by increases in summer precipitation and vis versa.

We agree with the reviewer that the Spannagel $\delta_{18}\text{O}$ record does not allow for the reconstruction of seasonal precipitation amounts or changes, but only to infer if relative portions have changed. For this reason we decided to use the more broadly based reconstructed of net precipitation changes from Routson et al. (2019) instead of the Spannagel $\delta_{18}\text{O}$ record. This reconstruction has also the advantage that it uses numerous precipitation reconstructions from across the Northern Hemisphere mid-latitudes and is therefore likely to be far more representative and robust than a single speleothem $\delta_{18}\text{O}$ record from Spannagel Cave.

Comment: The proposed forcing mechanism, (reducing the LTG) would probably result in a weaker average jet stream rather than a marked shift in position. If anything, there might be a somewhat equatorial shift in the circulation.

We agree with the reviewer and revised this paragraph. In the revised manuscript we cite the relevant publications that support the conclusion of a weaker jet stream when LTGs between the low- and the high-latitudes are reduced (e.g. Routson et al., 2019).

Line 147-148. The compilation in Ref 3 is assessing total effective precipitation, blending seasons, in a broad region where precipitation is primarily influenced by extratropical cyclones. Thus, it remains unclear how the general Holocene pattern of mid latitude effective precipitation (low early Holocene toward higher late Holocene) is reflected seasonally.

We agree with the reviewer that this was not well explained. We revised the relevant text section in manuscript accordingly and shortly summarize the discussion of Routson et al. on the seasonal implications of their reconstruction (Lines 149-151).

Lines 222-246. Connecting solar variability to forcing the high frequency variability in these records is unconvincing. Tuning and wiggle matching records within age uncertainties will always result in higher correlations and interesting patterns to discuss. It's surprising that even higher correlations were not achieved through this method.

We agree with the reviewer that this conclusion was weak and have deleted it from the manuscript to re-focus on the much more robust orbital-scale changes in the hydroclimate reconstructions.

Reviewer #2 (Remarks to the Author):

The manuscript "Persistent inter-hemispheric synchronicity of Holocene precipitation anomalies in Europe, East Africa and South America controlled by Earth's latitudinal temperature gradients" by Deininger et al. is an interesting and important contribution to the understanding of the various monsoon systems and the impact of insolation gradients on precipitation across the hemispheres. It convincingly shows that proxy paleohydrological records from three continents located at the NH, Tropics and SH. The fact that tropical climate systems (and proxy records thereof) can be influenced by what is happening in the extra-tropics is something that has not been very well understood and this manuscript shades some light on this, which in my opinion is an excellent contribution.

However, the authors need to consider the following issues concerning the data, discussion and conclusions in view of improving the quality of the manuscript.

General Comments:

1. The authors did not make it clear why these three sites were particularly selected for comparison. This leads to several other questions: how comparable are the data? Do the different proxies really respond to the same climate forcing? At what resolution does the comparison make sense? Millennial? Centennial? This is particularly important when you compare the two speleothem records with the Lake Tana record... There seems to be some growing evidence that Lake records in the tropics may not be directly comparable with speleothem records particularly during variable climate (which is particularly true for the Late Holocene).

We agree with the reviewer. We revised the method section accordingly to make the new data selection as transparent as possible. Furthermore, in the revised manuscript we focus only on the orbital-scale changes in the hydroclimate reconstructions.

2. What was the reason for selecting Lake Tana record for the tropics? There are several other Holocene records from the tropics with probably more resolution than Lake Tana. And there is a more recent record/or interpretation of the lake record of Lake Tana which uses Ca/Ti ratio as a proxy for wetness/dryness (rainfall or hydrological proxy) than the absolute Ti content (Lamb et al. 2018; Scientific Reports, 8). I suggest that the authors refer to this record instead.

We agree with the reviewer that the reason for the selection was not clear. In the revised manuscript we use 25 hydroclimate records from across Africa, of which the Lake Tana Ti-concentration record is one. The selection of these records is explained in detail in the methods section. However, we did not use the Ca/Ti record for the following reasons: Lamb et al. (2018) argues that the Ca/Ti ratio is sensitive to evaporation of water from Lake Tana, in which the Ca-concentration increases when the lake level decreases during dryer periods. This requires that the lake level is influenced only by the inflow of water from rivers and the ‘outflow(flux)’ of water from Lake Tana occurs only via evaporation. However, the outflow of water from Lake Tana into the Blue Nile during the Holocene may cause non-linearity Ca/Ti-relationships with the effective water balance of Lake Tana. This is likely the reason why Lamb et al. (2018) used the Holocene Ca/Ti-ratios only as a baseline for a wet climate. However, we note that using the Ca/Ti-record instead of the Ti-concentration record would not have affected the results of the African composite record.

3. The authors seem unfamiliar with the literature on the Quaternary records of tropical Africa (see specific comments for some suggestions to improve this).

Specific comments:

Line 2 (and where ever it reappears in the manuscript with the same context): “eastern Africa” is the correct term if you refer to Ethiopia (where Lake Tana is located). “East Africa” is more a political/economic block to which only Kenya, Tanzania and Uganda belong. In fact, the region where the East African Monsoon System affects is “eastern Africa”, which includes also “East Africa”

We apologize for this inaccuracy. We revised the manuscript accordingly.

Lines 96-97: The papers cited are not the appropriate ones for describing the climate system in Ethiopia. I am not prescribing any specific papers but some suggestions are the following or any similar others on the climate systems in the horn of Africa/Eastern Africa: Segele et al., 2009; Journal of Climate, 22; Segele et al., 2009, International Journal of Climatology, 29, etc.

Lines 107-109: Detrital input could also be related to short but intense rainfall within a generally dry period. I think using an elemental ratio record (such as Ca/Ti) will be better suited than an absolute Ti content which is not easy to interpret under various hydrological conditions.

Please see our reply to this general comment.

Line 153 (Figure 2): The pictures are not easy to understand as there are several variables plotted (4 in panel b). It is better to organize them differently (or use very contrasting colours for the different proxy curves) in order to clearly see how the curves trend...Panel b is very busy and it is really difficult to see if there is correlation or not.

We agree with the reviewer that this figure was very ‘busy’. We revised the figure (now Figure 3) accordingly.

Lines 181-188: Show what is discussed here in the diagram (as panel c of figure 2, for instance) similar to the other comparisons.

We revised the figure as well as the text accordingly based on the new results.

Lines 212-213 (caption to Figure 3): Is there any particular reason to reverse only these? And it is not clear if the axis is different in panel a.

We have the feeling that the comparison to solar variations was a weak part of the manuscript. For this reason we deleted this part from the revised manuscript. Furthermore, we agree with the reviewer that the figure caption of Fig. 3 was not well explained.

Lines 219-220 (caption to Figure 3): same comment as above.

We agree. We deleted Figure 3 (see above).

Line 231: typo "Laka"

We deleted this part of the manuscript.

Lines 319-321: There are several works which show that the ITCZ was high up in the North in Africa during the Early to Middle Holocene [It is recommended to cite some previous works which show this was the case, as I do not believe this conclusion is new] but it is interesting to see that the same mechanism (that led to the ITCZ to hang up in the North) had more global inter-hemispheric reach and implication.

We included several new records from across Africa in our compilation of hydroclimate proxy records and comment on the similarity between changes in the African Monsoon and shifts in the mean position of the ITCZ. Please see also our general reply to all reviewers.

Lines 392-394: This is an important conclusion which will have a big societal implication. But the conclusion could be more justified if the authors tried to support this conclusion by what other works show (such as modeling works or even current trends from instrumental rainfall records).

We agree with the reviewer that this conclusion was possibly too far reaching. For this reason we modified this final conclusion (Lines 360-367 in the revised MS). We also added a sentence that the increasing atmospheric CO₂ concentration may cause non-linear processes, such as the predicted and observed widening of the Hadley cells.

Reviewer #3 (Remarks to the Author):

The authors analyze three records related to precipitation changes in Central Europe, East Africa and southeastern Brazil that display strong similarities over the Holocene. From those records, the authors argue that the changes in the North Atlantic storm tracks, the East African Monsoon System, and the South American Monsoon System are linked and controlled by latitudinal gradient of insolation and temperature.

The strong correlation between the three records is interesting and intriguing and the authors address fundamental questions about the dynamics of our climate. However, I cannot recommend the publication of the manuscript in its present form as the strong conclusions reached by the authors are not sufficiently justified on the basis of their results and of their discussion.

Two main points justify this recommendation.

1/ Three records only are selected to derive general conclusions about the climate dynamics at large-scale. However, there is nearly no discussion of the reason of the choice of those records, except that there are located in contrasted conditions. We know that precipitation records can display large variability, even at regional scale. A brief comparison with some European records is provided figure S4 but, intriguingly, the correlation between the European records seems lower than between the selected European record and the ones in

East Africa and Brazil. Before using those three records for the analyses, it is necessary to determine how they compare to other records in nearby regions. It would even be better to use a large dataset and see how this influences the proposed conclusions.

We agree with the reviewer. For this reason we use a larger hydroclimate dataset that we analyzed for coherent changes (see our general reply to all reviewers).

2/ The authors propose a conceptual model (summarized on Figure 1) to explain the observed variations in the three records, based on some general physical principles and existing literature but the reasoning leading to this model is speculative and the proposed hypothesis are not properly justified or tested. First, the discussion is a bit loose on the concepts themselves as well as on the season analyzed. Is the figure 1 representing a zonal mean for the whole Earth (while we know that different longitudes can behave quite differently) or just for the Atlantic Sector which is closer to the data selected? Is it a figure on annual mean or for a specific season?

We revised the figure caption of our conceptual figure and state that the picture is only representative for the Atlantic sector.

Several hypotheses have been proposed about circulation changes in the Holocene but alternative hypotheses are not discussed at all. A large part of the argumentation is based on ref 4 but not on other work on the subject. Specifically, the authors do not explain why their conceptual model explains better the records than alternative ones. They argue for a dynamic link between the three regions but a direct link to local insolation may also explain the recorded signal (at least in some regions, for instance in the regions of monsoon as argued in some studies, e.g. line 56)?

We included a paragraph on the conventional view of monsoon dynamics and how these are related to mechanisms that have been proposed to explain variations in the African and the South American Monsoons (Lines 206-228).

The link with the latitudinal gradient does not seem particularly convincing to me from Fig. 2. The differences before 7 ka might be due to the remnant ice sheets but they are quite large to be justified by this effect only. However, even after that, the maximum in gradient for Europe (Fig 2 a) occurs basically 1 ka later than the maximum in the precipitation records. Finally, all the records are relatively flat over the last 4000 years while the latitudinal gradient decreases clearly over this period. Without additional, more quantitative analyses, it is thus difficult to justify the strong conclusions reached by the authors.

In the revised manuscript we use a new compilation of hydroclimate proxy records from the realm of the African and the South American Monsoons to investigate coherent variations in the respective hydroclimates. In addition, we use the results obtained by Routson et al. (2019) for the Northern Hemisphere mid-latitudes to complement the monsoon data. The new composite records for the African and South American Monsoons show a strong relationship with inter-hemispheric insolation contrasts and are in agreement with theoretical predictions. The results provides now firm and robust new insights into Holocene precipitation changes in the Atlantic sector.

An additional point is that I found the discussion of the solar forcing also quite speculative and it misses one important point. The response to TSI should be different from the one to orbital forcing. In particular, the effect of solar forcing should be much more homogenous between the season and the two hemispheres. This point is nearly not discussed. Furthermore, the authors did not mention solar forcing in the abstract or in the introduction so it appears disconnected to the main theme of the paper.

We agree with the reviewer. For this reason we have deleted the relevant parts of the manuscript that referred to solar forcing. The revised manuscript now focuses only on the orbital-scale changes, which yielded more robust results.

The authors use a lot of acronyms with which the reader is likely not familiar (and are sometimes defined after they are used such as LIG in the abstract). This makes the paper sometimes difficult to follow.

We agree with the reviewer and reduced the number of acronyms to a minimum.

Reviewers' Comments:

Reviewer #1:

Remarks to the Author:

The revised manuscript by Dr. Deininger et al., is a notable improvement. The authors are doing important work, and they made a commendable effort to compile records to characterize the African and South American monsoon systems. They also present some interesting hypotheses. However, there are considerable problems with the manuscript, and I still cannot recommend it for publication.

One important problem with their thesis is as follows. Their hypothesis revolves around cross-equatorial insolation and temperature gradients driving changes in Holocene monsoon systems. They argue the African Monsoon increases when the Northern Hemisphere receives more insolation than the Southern Hemisphere in boreal summer, and vis versa for the South American Monsoon. This mechanism involves energy transport from the cool hemisphere to the warm hemisphere, enhancing monsoon energy. Thus, Northern Hemisphere and Southern Hemisphere monsoon systems are needed to test this proposed mechanism. Northern Hemisphere components of the African Monsoon system could suffice. However, the African Monsoon composite (which follows the Northern Hemisphere pattern) includes 16 (out of 25) records from the Southern Hemisphere. These records extend to 32°S. The authors highlight one of the Southern Hemisphere African Monsoon records (Orange River) as matching the South American composite and supporting their cross-equatorial or interhemispheric hypothesis. What about the 15 other Southern Hemisphere records in Africa? Those should match the South American monsoon as well, and should not be included in a composite representing a Northern Hemisphere pattern.

The evidence presented also doesn't contradict or refute the well supported traditional mechanism of monsoon variations on orbital timescales. Traditionally, enhanced summer insolation heats the land surface faster than the ocean, leading to enhanced land-sea thermal contrasts, and enhanced monsoon circulation. Are there differences in the timing of the new interhemispheric gradients versus direct insolation mechanisms? Both mechanisms could be occurring simultaneously. If so, is there evidence indicating your proposed mechanism is more important? Or occurring at all? The potential collinearity between land-sea thermal contrasts versus interhemispheric insolation contrasts make it difficult to assess if the proposed mechanism is important. A supplemental plot showing Holocene summer insolation for the monsoon regions (or monsoon region latitudes) compared with the interhemispheric insolation gradients for the same seasons (and the monsoon composites) might unravel the differences between direct insolation versus insolation gradients.

Next, the manuscript jumps between Monsoons, the Northern Hemisphere Jetstream, Holocene timescales, the Last Glacial, 120 thousand years, and 700 thousand years. I believe these transitions and connections are clear in the authors mind, but they result in discontinuity not well articulated or linked in the abstract or introduction. These systems and timescales could be connected, but the manuscript doesn't do a good job of tying the threads.

The analysis and discussion the Jetstream in southern Europe is confusing and apparently contradicts the associated plots. This discussion needs clarified and needs to be clearly linked to the story arc. See line specific (317-345) comment below.

Finally, the manuscript needs considerable work for clarity and writing. The writing includes many wordy statements and is often difficult to follow. Complex ideas need to be distilled into succinct sentences.

Line specific comments:

Line 159-160: An increase in the African monsoon occurs when the NH receives more insolation relative to the SH in boreal summer. But African Monsoon records are mostly in the Southern Hemisphere?

169-172. As written I don't think this sentence makes sense. Insolation didn't force changes in circulation to adjust poleward energy transport. Insolation changed. Temperature responded. Then circulation respond.

Figure 3. The errors are extremely small in plots c and d given the small sample sizes. Africa $n = 25$ and S. America $n = 12$. The large amount of noise inherent among hydroclimate records should result in very large uncertainties. For example, the Orange River record (included in the African Composite) has the opposite pattern as the composite, which should result in some amount of uncertainty. I would recommend (at minimum) applying a bootstrapped sampling with replacement approach. In Matlab see `datasample.mat` <https://www.mathworks.com/help/stats/datasample.html>, and/or `bootstrp.mat` <https://www.mathworks.com/help/stats/bootstrp.html>.

Figure 3: Flipping the directions (positive up versus down) of the latitudinal insolation (LIG) curves between c and d. Please make this obvious. I didn't notice this until the discussion section (line 235) when you state the LIG decreased in both seasons. Also make the differences among the LIG curves clearer without having to dig through the text by adding a legend or text next to the curves (e.g. Austral Summer Interhemispheric Insolation Gradient. Or something similar).

Figure 3. Can you explain why the South American monsoon begins to diverge from the insolation curve in the late Holocene?

All figures: In my opinion time should progress forward from left to right like reading a book. I realize there are different conventions here, which is confusing.

Line 238: delete "indeed"

Line 240-244. I'm confused. One record from south Africa (Orange River) matches the South American Monsoon, and you argue that this is because this record is in the South African Precipitation region. Was this record also included in the African Monsoon composite as indicated (one of 25 records listed in the supplemental data file)? What about all the other records right next to the Orange River? There are sites from 32N to 32S in the African Monsoon composite, which is traversing regions which, by the hypothesis presented, should behave in opposite directions.

Line 245: "which provokes in summary"? Reword. Extra words and phrases throughout the manuscript make it difficult to follow. Don't be fancy, but clear and concise. Clear writing reflects clear thinking.

Line 247: Reword to "Therefore, African Monsoon precipitation ..." or "Therefore, African precipitation..." or something similar...

Line 255. What times?

Line 257: Dramatically shifting timeframes here. Discussion of the last glacial should at least have their own paragraph if they belong in the paper at all. The evidence and arguments are focused on the Holocene, explicitly after 7 ka. Thus, it is not clear how inference to other intervals is relevant to the conversation without a substantial amount of additional evidence or framing.

Line 265: Since 10ka or 7ka as on line 77?

Line 290: Dramatically shifting timescales again...

Lines 317-345. The authors present an interesting regional analysis of hydroclimate differences between Northern and Southern Italy. 1) As presented, this topic feels tangential. The authors need to do a better job of illustrating how this regional finding is relevant to the arc of the story. Alternatively, stay focused on monsoons and interhemispheric energy transport. 2) I'm not following how the statement (lines 335-338) "The persistent change of European LTGs indicate that the mean position of the North Atlantic storm-tracks shifted equatorwards from the early to the late Holocene" reconciles with Figure 5 panels e and f. Am I missing something? Figure 5 shows Northern Italy got wetter, and southern Italy got drier. Doesn't this result indicate a poleward shift in the jet stream rather than an equatorward shift? More generally, the discussion presented between lines 334 and 345 is difficult to follow.

Reviewer #2:

Remarks to the Author:

This revised manuscript is substantially improved. My major concern in the previous version was the number of records (too few) used for arriving at the conclusions and it was not clear how these records were selected. In the current version, the authors substantially increased the number of records (with a wider spatial coverage) and were able to show that their conclusions are justified. I will be happy to accept this version, except the following minor issues have to be addressed:

Title: Is very long and complicated. Inter-hemispheric and "Europe, Africa and South America" imply the same thing, and one of these phrases could be dropped.

Line 176-172: This paragraph could be included in the discussion section and further elaborated.

Line 246: "summary"? do you mean summer?

Reviewer #3:

Remarks to the Author:

Compared to the submitted version, the authors have well addressed my first concern about the selection of a few data only. They analyze in this new version a much larger data set and those additional series confirm the trends deduced from the submitted version. The signal derived from the data appears thus much more robust.

However, I still have concerns about the argumentation justifying the conceptual framework. The authors argue that the trends over the Holocene can be explained by the changes in latitudinal temperature gradients. The presentation of the mechanism appears reasonable. For the mid-latitudes, the mechanism is to the first order similar to the one presented in reference 5 (which is often cited, in the text and in the rebuttal letter). For monsoon regions, the dominant role of temperature gradients proposed in the manuscript contrasts with a more classical explanation that considers that local insolation has the largest effect (see line 54-55 of the manuscript for instance).

This dominant role of local insolation could also be justified using theoretical arguments. The choice between the two theories (local insolation or insolation gradient) should be based on their ability to fit

with the data. The authors show a reasonable fit between the trends in the latitudinal gradients of temperature (or insolation) and the precipitation trends. They discuss the potential role of local insolation and then discard it on the basis of this discussion. However, the authors do not show whether the local insolation trend may fit as well to the data, in the equivalent of Fig. 3 for instance. The authors seem also a bit confused at a later stage of the paper about the respective role of local insolation and insolation gradient as they mention line 273-278 'However, if the change in the cross-equatorial energy flux cannot fully compensate the 'available' excess energy in the warmer tropical hemisphere, the tropical summer hemisphere will warm. This will likely strengthen the tropical atmospheric circulation especially over the continents where the heat capacity is lower compared to the ocean. This process is similar to the traditional insolation-based mechanism in strengthening the land-sea thermal gradient when TOA insolation increases'. If I understand well, it is mentioned in this sentence that local insolation can also play a role but, then, how can we quantify the contribution of each process? There also seems to have a contradiction with lines 222-224 'However, the dynamic aspect of the atmospheric circulation would be ignored if the reconstructed hydroclimate composite records are compared to changes of TOA insolation values only' while the text quoted above discusses the classical strengthening of the circulation related to local warmings.

Of course, in the real world, local insolation and insolation gradient are not independent but I do not consider that the qualitative arguments presented in the manuscript are strong enough to justify the conclusion '... the precipitation anomalies in Europe, Africa and South America controlled by Earth's latitudinal temperature gradients' as stated in the title.

Specific comments

Lines 39-42. In the core of the paper (line 368), the authors correctly argue that the future changes will likely be very different from the ones occurring in the Holocene. I would thus not include a reference to this link between past and future changes in the abstract.

Line 72. What is meant by 'Our independent reconstructions'? I would not use 'our' here for all the reconstructions.

Line 118. The fit between precipitation and insolation gradient seems to be lost after 3ka BP. Is it a sign that it is not the insolation gradient that controls the precipitation but another process (that correlates with insolation gradient earlier)?

Line 207. I do not understand what is meant here by 'moisture concentrations'.

Lines 255-256 'At times when the NH extra-tropics were warmer than the tropics'. Is it the mean state or the anomaly?

Lines 287-293. Here, like for the initial submission, why focusing on only a few records and periods? If the authors want to discuss the general role of insolation gradient in climate dynamics, they have to analyze precisely all the periods of interest, not just using time series that seems to confirm their hypothesis.

Lines 317-345. I have several problems with the discussion of the changes in Europe in this section. If I understand well, the authors argue that, in addition to the modification described in ref.5, the position of the storm tracks in Europe has been shifted in response to changes in latitudinal temperature gradient. The authors link the changes in latitudinal gradient with the NAO using a concept based on modern data (ref 58, 59). However, when analyzing interannual variability, the changes in latitudinal gradients are a consequence of the shift in winds, not a cause. By contrast, in

the theory proposed by the authors for the Holocene, the changes in temperature, driven by insolation changes, are at the origin of the wind changes. Therefore, I do not understand how the NAO concepts fits in the argumentation proposed in the paper. Specifically, line 334-335 'Using the present-day relationship between European LTGs and the mean latitudinal position of storm tracks across Europe ...' is not valid for me as the European LTG can be simply controlled by the latitudinal gradients of insolation. This is actually the hypothesis that seems to be applied in the other parts of the paper where latitudinal gradient of temperature or of insolation are supposed to be strongly linked (see for instance figure 3). Additionally, the shift in precipitation are only shown for Southern Italy (+ two references). This is not enough to confirm that this corresponds to a shift in the storm tracks.

Dear Reviewers,

Thank you very much for your rigorous review, your comments and ideas. These are greatly
appreciated and led to an improved manuscript and strengthened our main findings.

(on behalf of all authors)

Referee comments

Authors response

**Reviews**

**Reviewer #1 (Remarks to the Author):**

The revised manuscript by Dr. Deininger et al., is a notable improvement. The authors are
doing important work, and they made a commendable effort to compile records to
characterize the African and South American monsoon systems. They also present some
interesting hypotheses. However, there are considerable problems with the manuscript, and I
still cannot recommend it for publication.

One important problem with their thesis is as follows. Their hypothesis revolves around cross-
equatorial insolation and temperature gradients driving changes in Holocene monsoon
systems. They argue the African Monsoon increases when the Northern Hemisphere receives
more insolation than the Southern Hemisphere in boreal summer, and vis versa for the South
American Monsoon. This mechanism involves energy transport from the cool hemisphere to
the warm hemisphere, enhancing monsoon energy. Thus, Northern Hemisphere and Southern
Hemisphere monsoon systems are needed to test this proposed mechanism. Northern
Hemisphere components of the African Monsoon system could suffice. However, the African
Monsoon composite (which follows the Northern Hemisphere pattern) includes 16 (out of 25)
records from the Southern Hemisphere. These records extend to 32°S. The authors highlight
one of the Southern Hemisphere African Monsoon records (Orange River) as matching the
South American composite and supporting their cross-equatorial or interhemispheric
hypothesis. What about the 15 other Southern Hemisphere records in Africa? Those should
match the South American monsoon as well, and should not be included in a composite
representing a Northern Hemisphere pattern.

To better constrain the regional precipitation changes in Africa and the link to the South
American Monsoon, we calculated additional regional composite records. The African
regions were divided by their latitude including the Northern Hemisphere, the tropical belt of
the Southern Hemisphere (0° to 5.6 °S) (Supplementary Figure 1) and southern Africa (Figure
3). Because of opposite precipitation trends in southeastern and southwestern Africa, we
added two more composite records for these two regions in southern Africa as well (Figure 4).

The results of the regional composite records for the Northern Hemisphere and the Southern
Hemisphere tropical belt (Supplementary Figure 1) indicate similar precipitation trends,
showing a wet early Holocene and a drier late Holocene. Therefore, there is no seasonal effect

observed for these two regions, despite the fact that they are located in the Northern and
Southern Hemisphere. This is probably explained by the proximity of the Southern
Hemisphere records within the tropical belt to the Equator (5.6°S). A physical reason for this
observation is that it's not the geographical Equator is important, but rather the energy-flux
equator, which appears to be deflected into the Southern Hemisphere, allowing the Northern
Hemisphere summer forcing exert its influence also in the tropical regions of the Southern
Hemisphere. For this reason, we used the records of both regions to calculate a composite
record for the African Monsoon.

However, the composite record for the summer rainfall zone of southern Africa clearly shows
an opposite trend compared to the African Monsoon, but a similar Holocene trend as the
South American Monsoon. We argue that this is caused by the seasonal insolation forcing of
the African Monsoon and precipitation in southeastern Africa, which peak during boreal
summer and austral summer respectively.

Therefore, opposite precipitation trends are observed for regions where precipitation is under
the influence of boreal and austral summers, but the division of these regions does not depend
strictly on the geographical equator. To better resolve the precipitation pattern and to
investigate the regions which are under influence of boreal and austral summer, more high-
resolution hydroclimate reconstructions are required, particular in the region south of 5.6 °S.

The evidence presented also doesn't contradict or refute the well supported traditional
mechanism of monsoon variations on orbital timescales. Traditionally, enhanced summer
insolation heats the land surface faster than the ocean, leading to enhanced land-sea thermal
contrasts, and enhanced monsoon circulation. Are there differences in the timing of the new
interhemispheric gradients versus direct insolation mechanisms? Both mechanisms could be
occurring simultaneously. If so, is there evidence indicating your proposed mechanism is
more important? Or occurring at all? The potential collinearity between land-sea thermal
contrasts versus interhemispheric insolation contrasts make it difficult to assess if the
proposed mechanism is important. A supplemental plot showing Holocene summer insolation
for the monsoon regions (or monsoon region latitudes) compared with the interhemispheric
insolation gradients for the same seasons (and the monsoon composites) might unravel the
differences between direct insolation versus insolation gradients.

We added a new supplementary figure that compares the changes in the composite
hydroclimate records with changes in latitudinal summer insolation gradients and local
summer insolation. The changes in local summer insolation could indeed also explain the
observed hydroclimate changes linked to the African and South American Monsoons via the
land-ocean mechanism, indicating that the monsoons are stronger when the respective
summer insolation is high. This is overall not surprising considering that local summer
insolation is conventionally used to explain past monsoon changes via the land-ocean
mechanism. Furthermore, because the imposed precipitation changes of our insolation
gradient-based energy-budget framework and the insolation-based land-ocean mechanism
result in similar precipitation changes during the Holocene it is also not possible to tear these
two mechanisms apart. However, recent advances in the understanding of the tropical
circulation, and monsoon systems in particular, highlight the fundamental role of the tropical
atmospheric circulation as part of the global atmospheric energy conveyor belt (Biasutti et al.,
2018). Furthermore, this study argues that the conventional land-ocean mechanism is too
simplistic and indicates shortcomings of traditional land-ocean mechanism, for example that
the Monsoon lands are hottest before the monsoons start, but the circulation is strongest in
late summer, when the temperature contrast is smaller (because of increased rain and

cloudiness). Therefore, there is an urgent need for a revised conceptual framework for the
interpretation of past monsoon changes, that is based on Earth's energy budget (Biasutti et al.,
2018). Our new results demonstrate that such an energy-budget framework of the tropical
atmospheric circulation should be based on the differential heating of the Northern and
Southern Hemispheres, expressed by the inter-hemispheric insolation contrast and latitudinal
temperature gradients and yields not only a coherent concept that links north-south migrations
of the ITCZ with monsoon changes but also links changes in tropical atmospheric circulation
to extra-tropical atmospheric circulation.

Next, the manuscript jumps between Monsoons, the Northern Hemisphere Jetstream,
Holocene time-scales, the Last Glacial, 120 thousand years, and 700 thousand years. I believe
these transitions and connections are clear in the authors' mind, but they result in discontinuity
not well articulated or linked in the abstract or introduction. These systems and timescales
could be connected, but the manuscript doesn't do a good job of tying the threads.

To keep the focus on the Holocene changes only, we deleted the passages of text that jumped
between the different periods.

The analysis and discussion of the Jetstream in southern Europe is confusing and apparently
contradicts the associated plots. This discussion needs clarification and needs to be clearly linked
to the story arc. See line specific (317-345) comment below.

To keep the focus on the composite hydroclimate records, we decided to delete this part of the
manuscript. This has allowed us to better work out the observations and implications of the
synchronous precipitation changes associated with the Northern Hemisphere storm tracks and
the African and South American Monsoons.

Finally, the manuscript needs considerable work for clarity and writing. The writing includes
many wordy statements and is often difficult to follow. Complex ideas need to be distilled
into succinct sentences.

We thoroughly revised the manuscript to clearly work out our idea of an energy-budget
framework.

Line specific comments:

Line 159-160: An increase in the African monsoon occurs when the NH receives more
insolation relative to the SH in boreal summer. But African Monsoon records are mostly in
the Southern Hemisphere?

See our response to your first main comment.

169-172. As written I don't think this sentence makes sense. Insolation didn't force changes in
circulation to adjust poleward energy transport. Insolation changed. Temperature responded.
Then circulation responds.

This is true, we changed the sentence to: "..., we hypothesize that changing Holocene inter-
hemispheric and hemispheric latitudinal insolation gradients controlled the latitudinal
temperature gradients between the low- and high-latitudes, in turn forcing changes in
atmospheric circulation to adjust the poleward energy transport."

Figure 3. The errors are extremely small in plots c and d given the small sample sizes. Africa
$n = 25$ and S. America $n = 12$. The large amount of noise inherent among hydroclimate
records should result in very large uncertainties. For example, the Orange River record
(included in the African Composite) has the opposite pattern as the composite, which should
result in some amount of uncertainty. I would recommend (at minimum) applying a
bootstrapped sampling with replacement approach. In Matlab see `datasample.mat` , and/or
`bootstrp.mat` <https://www.mathworks.com/help/stats/bootstrp.html>.

We indeed calculated individual standard errors from moving-block bootstrap resampling
(Mudelsee, 2014) and obtained the weighted average (shown in the figure) with correctly
estimated external error bands (calculated from the individual records). We firmly believe that
"our error bars" are correct.

Figure 3: Flipping the directions (positive up versus down) of the latitudinal insolation (LIG)
curves between c and d. Please make this obvious. I didn't notice this until the discussion
section (line 235) when you state the LIG decreased in both seasons. Also make the
differences among the LIG curves clearer without having to dig through the text by adding a
legend or text next to the curves (e.g. Austral Summer Interhemispheric Insolation Gradient.
Or something similar).

We revised the figures and indicate the direction of increasing insolation contrasts with
arrows and don't abbreviate the LIGs anymore, but use full text (e.g. JJA insolation contrast
$(30^{\circ}\text{N} \text{ minus } 30^{\circ}\text{S} \text{ in } \text{W m}^{-2})$).

Figure 3. Can you explain why the South American monsoon begins to diverge from the
insolation curve in the late Holocene?

This effect was caused by the Paraiso record, which shows a pronounced decrease in
precipitation amounts since the mid-Holocene (Figure 5). After a thorough discussion of this
observation among the group of co-authors who are experts for the South American Monsoon,
we decided to omit the Paraiso record for the calculation of the composite South American
Monsoon record. This is because precipitation at the Paraiso cave falls not only during austral
summer when the South American Monsoon is active, but during the entire year and in
particular during boreal summer (Figure 2). Therefore, the precipitation changes that are
inferred from the Paraiso record are not only related to variations of the South American
Monsoon but also to migrations of the ITCZ and changes in precipitation amounts during
boreal summer. We argue in the manuscript that the observed Holocene changes in
precipitation amounts at Paraiso are caused by seasonality changes and that the pronounced
decrease in precipitation amounts is caused by a decrease in boreal summer precipitation.

All figures: In my opinion time should progress forward from left to right like reading a book.
I realize there are different conventions here, which is confusing.

Line 238: delete "indeed"

Revised.

Line 240-244. I'm confused. One record from south Africa (Orange River) matches the South
American Monsoon, and you argue that this is because this record is in the South African
Precipitation region. Was this record also included in the African Monsoon composite as
indicated (one of 25 records listed in the supplemental data file)? What about all the other

records right next to the Orange River? There are sites from 32N to 32S in the African
Monsoon composite, which is traversing regions which, by the hypothesis presented, should
behave in opposite directions.

See our response to your first main comment.

Line 245: “which provokes in summary”? Reword. Extra words and phrases throughout the
manuscript make it difficult to follow. Don’t be fancy, but clear and concise. Clear writing
reflects clear thinking.

Revised.

Line 247: Reword to “Therefore, African Monsoon precipitation ...” or “Therefore, African
precipitation...” or something similar...

Revised.

Line 255. What times?

We revised the sentence to: “During the early Holocene, from 10 ka BP to 6 ka BP, when the
NH extra-tropics were warmer than the tropics, ...”

Line 257: Dramatically shifting timeframes here. Discussion of the last glacial should at least
have their own paragraph if they belong in the paper at all. The evidence and arguments are
focused on the Holocene, explicitly after 7 ka. Thus, it is not clear how inference to other
intervals is relevant to the conversation without a substantial amount of additional evidence or
framing.

To keep the focus on the Holocene changes only, we deleted the passages of text that jumped
between the different periods. Nevertheless, these comparisons did demonstrate that our
energy-budget framework could also explain past monsoon changes on much longer time
scales.

Line 265: Since 10ka or 7ka as on line 77?

In fact, since 10 ka, because the effects of the remnant ice sheets are only inferred for the
Northern Hemisphere mid-latitude atmospheric circulation.

Line 290: Dramatically shifting timescales again...

To keep the focus on the Holocene changes only, we deleted the passages of text that jumped
between the different periods. Nevertheless, these comparisons did demonstrate that our
energy-budget framework could also explain past monsoon changes on much longer time
scales.

Lines 317-345. The authors present an interesting regional analysis of hydroclimate
differences between Northern and Southern Italy. 1) As presented, this topic feels tangential.
The authors need to do a better job of illustrating how this regional finding is relevant to the
arc of the story. Alternatively, stay focused on monsoons and interhemispheric energy
transport. 2) I’m not following how the statement (lines 335-338) “The persistent change of
European LTGs indicate that the mean position of the North Atlantic storm-tracks shifted

equatorwards from the early to the late Holocene” reconciles with Figure 5 panels e and f. Am
I missing something? Figure 5 shows Northern Italy got wetter, and southern Italy got drier.
Doesn’t this result indicate a poleward shift in the jet stream rather than an equatorward shift?
More generally, the discussion presented between lines 334 and 345 is difficult to follow.

To keep the focus on the composite hydroclimate records, we decided to delete this part of the
manuscript. This has allowed us better work out the observations and implications of the
synchronous precipitation changes associated with the Northern Hemisphere storm tracks and
the African and South American Monsoons.

Reviewer #2 (Remarks to the Author):

This revised manuscript is substantially improved. My major concern in the previous version was the number of records (too few) used for arriving at the conclusions and it was not clear how these records were selected. In the current version, the authors substantially increased the number of records (with a wider spatial coverage) and were able to show that their conclusions are justified. I will be happy to accept this version, except the following minor issues have to be addressed:

Title: Is very long and complicated. Inter-hemispheric and "Europe, Africa and South America" imply the same thing, and one of these phrases could be dropped.

Revised.

Line 176-172: This paragraph could be included in the discussion section and further elaborated.

Revised. We start the discussion with this paragraph.

Line 246: "summary"? do you mean summer?

Reviewer #3 (Remarks to the Author):

Compared to the submitted version, the authors have well addressed my first concern about the selection of a few data only. They analyze in this new version a much larger data set and those additional series confirm the trends deduced from the submitted version. The signal derived from the data appears thus much more robust.

However, I still have concerns about the argumentation justifying the conceptual framework. The authors argue that the trends over the Holocene can be explained by the changes in latitudinal temperature gradients. The presentation of the mechanism appears reasonable. For the mid-latitudes, the mechanism is to the first order similar to the one presented in reference 5 (which is often cited, in the text and in the rebuttal letter). For monsoon regions, the dominant role of temperature gradients proposed in the manuscript contrasts with a more classical explanation that considers that local insolation has the largest effect (see line 54-55 of the manuscript for instance).

This dominant role of local insolation could also be justified using theoretical arguments. The choice between the two theories (local insolation or insolation gradient) should be based on their ability to fit with the data. The authors show a reasonable fit between the trends in the latitudinal gradients of temperature (or insolation) and the precipitation trends. They discuss the potential role of local insolation and then discard it on the basis of this discussion. However, the authors do not show whether the local insolation trend may fit as well to the data, in the equivalent of Fig. 3 for instance. The authors seem also a bit confused at a later stage of the paper about the respective role of local insolation and insolation gradient as they mention line 273-278

‘However, if the change in the cross-equatorial energy flux cannot fully compensate the ‘available’ excess energy in the warmer tropical hemisphere, the tropical summer hemisphere will warm. This will likely strengthen the tropical atmospheric circulation especially over the continents where the heat capacity is lower compared to the ocean. This process is similar to the traditional insolation-based mechanism in strengthening the land-sea thermal gradient when TOA insolation increases’.

If I understand well, it is mentioned in this sentence that local insolation can also play a role but, then, how can we quantify the contribution of each process? There also seems to have a contradiction with lines 222-224

‘However, the dynamic aspect of the atmospheric circulation would be ignored if the reconstructed hydroclimate composite records are compared to changes of TOA insolation values only’

while the text quoted above discusses the classical strengthening of the circulation related to local warmings.

Of course, in the real world, local insolation and insolation gradient are not independent but I do not consider that the qualitative arguments presented in the manuscript are strong enough to justify the conclusion ‘... the precipitation anomalies in Europe, Africa and South America controlled by Earth’s latitudinal temperature gradients ‘ as stated in the title.

We added a new supplementary figure that compares the changes in the composite
hydroclimate records with changes in latitudinal summer insolation gradients and local
summer insolation. The changes in local summer insolation could indeed also explain the
observed hydroclimate changes linked to the African and South American Monsoons via the
land-ocean mechanism, indicating that the monsoons are stronger when the respective
summer insolation is high. This is overall not surprising considering that local summer
insolation is conventionally used to explain past monsoon changes via the land-ocean
mechanism. Furthermore, because the imposed precipitation changes of our insolation
gradient-based energy-budget framework and the insolation-based land-ocean mechanism
result in similar precipitation changes during the Holocene it is also not possible to tear these
two mechanisms apart. However, recent advances in the understanding of the tropical
circulation, and monsoon systems in particular, highlight the fundamental role of the tropical
atmospheric circulation as part of the global atmospheric energy conveyor belt (Biasutti et al.,
2018). Furthermore, this study argues that the conventionally land-ocean mechanism is too
simplistic and indicates shortcomings of traditional land-ocean mechanism, for example that
the Monsoon lands are hottest before the monsoons start, but the circulation is strongest in
late summer, when the temperature contrast is smaller (because of increased rain and
cloudiness). Therefore, there is an urgent need for a revised conceptual framework for the
interpretation of past monsoon changes, that is based on Earth's energy budget (Biasutti et al.,
2018). Our new results demonstrate that such an energy-budget framework of the tropical
atmospheric circulation should be based on the differential heating of the Northern and
Southern Hemispheres, expressed by the inter-hemispheric insolation contrast and latitudinal
temperature gradients and yields not only a coherent concept that links north-south migrations
of the ITCZ with monsoon changes but also links changes in tropical atmospheric circulation
to extra-tropical atmospheric circulation.

Specific comments

Lines 39-42. In the core of the paper (line 368), the authors correctly argue that the future
changes will likely be very different from the ones occurring in the Holocene. I would thus
not include a reference to this link between past and future changes in the abstract.

We believe that this is an important implication of our results and kept this statement in the
abstract. However, we make clear in the summary that different mechanism may limit the
reliability of projections based on the 'early and mid-Holocene' analogue: "It should be
acknowledged that the rapidly increasing atmospheric CO₂ concentrations at present may
give rise to non-stationary effects, such as the widening of the Hadley cells^{7,8,64,65}, which in
turn may limit the reliability of projections based on the 'early and mid-Holocene' analogue."

Line 72. What is meant by 'Our independent reconstructions'? I would not use 'our' here for
all the reconstructions.

Revised.

Line 118. The fit between precipitation and insolation gradient seems to be lost after 3ka BP.
Is it a sign that it is not the insolation gradient that controls the precipitation but another
process (that correlates with insolation gradient earlier)?

This effect was caused by the Paraiso record, which shows a pronounced decrease in
precipitation amounts since the mid-Holocene (Figure 5). After a thoroughly discussion of
this observation among the group of co-authors who are experts for the South American

Monsoon, we decided to omit the Paraiso record for the calculation of the composite South
American Monsoon record. This is because precipitation at the Paraiso cave falls not only
during austral summer when the South American Monsoon is active, but during the entire
458 year and in particular during boreal summer (Figure 2). Therefore, the precipitation changes
that are inferred from the Paraiso record are not only related to variations of the South
American Monsoon but also to migrations of the ITCZ and changes in precipitation amounts
during boreal summer. We argue in the manuscript that the observed Holocene changes in
precipitation amounts at Paraiso are caused by seasonality changes and that the pronounced
decrease in precipitation amounts is caused by a decrease in boreal summer precipitation.

Line 207. I do not understand what is meant here by 'moisture concentrations'.

We revised this sentence and refer to the strengthening of the monsoon circulation only in the
revised manuscript.

Lines 255-256 'At times when the NH extra-tropics were warmer than the tropics'. Is it the
mean state or the anomaly?

It is the mean state.

Lines 287-293. Here, like for the initial submission, why focusing on only a few records and
periods? If the authors want to discuss the general role of insolation gradient in climate
dynamics, they have to analyze precisely all the periods of interest, not just using time series
that seems to confirm their hypothesis.

To keep the focus on the Holocene changes only, we deleted the passages of text that jumped
between the different periods. Nevertheless, these comparisons did demonstrate that our
energy-budget framework could also explain past monsoon changes on much longer time
scales.

Lines 317-345. I have several problems with the discussion of the changes in Europe in this
section. If I understand well, the authors argue that, in addition to the modification described
in ref.5, the position of the storm tracks in Europe has been shifted in response to changes in
latitudinal temperature gradient. The authors link the changes in latitudinal gradient with the
NAO using a concept based on modern data (ref 58, 59). However, when analyzing
interannual variability, the changes in latitudinal gradients are a consequence of the shift in
winds, not a cause. By contrast, in the theory proposed by the authors for the Holocene, the
changes in temperature, driven by insolation changes, are at the origin of the wind changes.
Therefore, I do not understand how the NAO concepts fits in the argumentation proposed in
the paper. Specifically, line 334-335 'Using the present-day relationship between European
LTGs and the mean latitudinal position of storm tracks across Europe
...' is not valid for me as the European LTG can be simply controlled by the latitudinal
gradients of insolation. This is actually the hypothesis that seems to be applied in the other
parts of the paper where latitudinal gradient of temperature or of insolation are supposed to be
strongly linked (see for instance figure 3). Additionally, the shift in precipitation are only
shown for Southern Italy (+ two references). This is not enough to confirm that this
corresponds to a shift in the storm tracks.

To keep the focus on the composite hydroclimate records, we decided to delete this part of the
manuscript. This has allowed us better work out the observations and implications of the

synchronous precipitation changes associated with the Northern Hemisphere storm tracks and
the African and South American Monsoons.

Reviewers' Comments:

Reviewer #1:

Remarks to the Author:

The authors have considerably improved their manuscript, and I appreciate their willingness to work through a long and challenging review process. I agree with the authors that the conceptual framework of past monsoons needs improved and expanded. The authors have finally distilled their argument into a compelling hypothesis largely supported by the evidence. A couple minor comments follow:

Line 41 delete second use of 'weakening' in sentence

Line 87-91: This sentence is difficult to track. Maybe something along the lines of:

"We use Holocene hydroclimate reconstructions from African monsoon, South American Monsoon, and Northern Hemisphere mid-latitude regions to characterize a synchronous response to changing latitudinal insolation and temperature gradients"

Lines 404-407: I don't recommend ending the paper with a well referenced caveat. The caveat is important, but not what people should leave with. A condensed version of the caveat might fit on line 397 with something similar to the following:

"...exert important controls on tropical Hadley circulation. Recent atmospheric CO₂ increases may cause non-stationary effects (e.g. refs 5,6,44,45), that could limit the reliability of projections based on an early- to mid-Holocene analogue. Nonetheless, our study emphasizes the importance of...."

Reviewer #3:

Remarks to the Author:

The new version of the manuscript has addressed my main concerns as it is more focused on the key results and presents in a more balanced way the proposed hypothesis compared to the potential role of local insolation. One may consider that the authors still overemphasize the strength of their arguments in favor of the role of latitudinal gradients and their conclusions but I can live with that as the readers can have their personal opinion from the information given.

However, I still find that it is not adequate to have 7 lines in the abstract devoted to speculations about future changes while this is not the subject of the paper. It is perfectly fine to have some perspectives at the end of the manuscript as proposed but the main focus of the abstract should be the results of the study itself.

More generally, I found the argumentation sometimes difficult to follow with many repetitions. In particular, the first paragraph of the Discussion section is very long and is devoted to many issues. Separating it in shorter, more focused paragraphs would be clearer for me. Additionally, I have trouble to see the added value of the section 'The development of an energy-budget framework'. It seems for me that nearly all the information has been given before and this section is mainly a repetition with no new argument or concept.

On a very specific point, line 188, does the authors really mean that the extra-tropics are warmer than tropical regions, i.e. that the temperature at 40°N is higher than at 20°N ? This is not the way I understand Figure 3.

Referee comment.

Authors reply.

Reviewer #1 (Remarks to the Author):

The authors have considerably improved their manuscript, and I appreciate their willingness to work through a long and challenging review process. I agree with the authors that the conceptual framework of past monsoons needs improved and expanded. The authors have finally distilled their argument into a compelling hypothesis largely supported by the evidence. A couple minor comments follow:

Line 41 delete second use of 'weakening' in sentence

Revised. We deleted the second use of 'weakening'.

Line 87-91: This sentence is difficult to track. Maybe something along the lines of:

"We use Holocene hydroclimate reconstructions from African monsoon, South American Monsoon, and Northern Hemisphere mid-latitude regions to characterize a synchronous response to changing latitudinal insolation and temperature gradients"

Revised. We adopted the suggested sentence of the referee.

Lines 404-407: I don't recommend ending the paper with a well referenced caveat. The caveat is important, but not what people should leave with. A condensed version of the caveat might fit on line 397 with something similar to the following:

"...exert important controls on tropical Hadley circulation. Recent atmospheric CO₂ increases may cause non-stationary effects (e.g. refs 5,6,44,45), that could limit the reliability of projections based on an early- to mid-Holocene analogue. Nonetheless, our study emphasizes the importance of...."

Revised. We changed the last paragraph as suggested by referee:

Reviewer #3 (Remarks to the Author):

The new version of the manuscript has addressed my main concerns as it is more focused on the key results and presents in a more balanced way the proposed hypothesis compared to the potential role of local insolation. One may consider that the authors still overemphasize the strength of their arguments in favor of the role of latitudinal gradients and their conclusions but I can live with that as the readers can have their personal opinion from the information given.

However, I still find that it is not adequate to have 7 lines in the abstract devoted to speculations about future changes while this is not the subject of the paper. It is perfectly fine to have some perspectives at the end of the manuscript as proposed but the main focus of the abstract should be the results of the study itself.

We revised this part of the abstract and deleted the projection of future climate changes based on the Holocene analogue.

More generally, I found the argumentation sometimes difficult to follow with many repetitions. In particular, the first paragraph of the Discussion section is very long and is devoted to many issues. Separating it in shorter, more focused paragraphs would be clearer for me. Additionally, I have trouble to see the added value of the section 'The development of an energy-budget framework'. It seems for me that nearly all the information has been given before and this section is mainly a repetition with no new argument or concept.

We thoroughly revised the discussion of the manuscript, looking particularly for repetitions. In detail, we split the section about the local precipitation response to the monsoon changes. One part, in which the precipitation patterns are described is copied in the results section and the second part is copied at the beginning of the discussion.

On a very specific point, line 188, does the authors really mean that the extra-tropics are warmer than tropical regions, i.e. that the temperature at 40°N is higher than at 20°N ? This is not the way I understand Figure 3.

No, that was not what we meant. We wanted to state the Northern Hemisphere extra-tropics warmed more than the tropical regions. We changed the sentence as follows: “Importantly, when the northern latitudes received more insolation relative to the southern latitudes (Fig. 3c,e), the latitudinal temperature gradient within the Northern Hemisphere was weaker, because the Northern Hemisphere extra-tropics are warmed more (Arctic amplification) than tropical regions^{3,18} (Fig. 3b).”